# LMCSleepNet: A Lightweight Multi-Channel Sleep Staging Model Based on Wavelet Transform and Muli-Scale Convolutions

**DOI:** 10.3390/s25196065

**Published:** 2025-10-02

**Authors:** Jiayi Yang, Yuanyuan Chen, Tingting Yu, Ying Zhang

**Affiliations:** 1College of Artificial Intelligence & Computer Science, Xi’an University of Science and Technology, Xi’an 710054, China; 2School of Electronic Information Engineering, Xi’an Technological University, Xi’an 710021, China

**Keywords:** automatic sleep stage classification, deep learning, dilated convolution, time–frequency feature, depth-separable convolution

## Abstract

Sleep staging is a crucial indicator for assessing sleep quality, which contributes to sleep monitoring and the diagnosis of sleep disorders. Although existing sleep staging methods achieve high classification performance, two major challenges remain: (1) the ability to effectively extract salient features from multi-channel sleep data remains limited; (2) excessive model parameters hinder efficiency improvements. To address these challenges, this work proposes a lightweight multi-channel sleep staging network (LMCSleepNet). LMCSleepNet is composed of four modules. The first module enhances frequency domain features through continuous wavelet transform. The second module extracts time–frequency features using multi-scale convolutions. The third module optimizes ResNet18 with depthwise separable convolutions to reduce parameters. The fourth module improves spatial correlation using the Convolutional Block Attention Module (CBAM). On the public datasets SleepEDF-20, SleepEDF-78, and LMCSleepNet, respectively, LMCSleepNet achieved classification accuracies of 88.2% (κ = 0.84, MF1 = 82.4%) and 84.1% (κ = 0.77, MF1 = 77.7%), while reducing model parameters to 1.49 M. Furthermore, experiments validated the influence of temporal sampling points in wavelet time–frequency maps on sleep classification performance (accuracy, Cohen’s kappa, and macro-average F1-score) and the influence of multi-scale dilated convolution module fusion methods on classification performance. LMCSleepNet is an efficient lightweight model for extracting and integrating multimodal features from multichannel Polysomnography (PSG) data, which facilitates its application in resource-constrained scenarios.

## 1. Introduction

Sleep is a life-sustaining activity. Sufficient sleep promotes both physical and mental health [1,2,3]. However, lack of sleep not only weakens memory [4] and cognitive functions [5], but also leads to increased reaction times, hormonal imbalances, depression, and hypertension [6,7,8]. Sleep staging is a crucial indicator for assessing sleep quality. Classifying the different stages of sleep provides deeper insights into sleep mechanisms. Moreover, sleep staging determines various conditions that may arise due to poor sleep quality.

Polysomnography (PSG) is a method to analyze sleep stage and sleep disorder diagnosis. The PSG includes electroencephalogram (EEG), electrooculogram (EOG), electromyogram (EMG), electrocardiogram (ECG), and blood oxygen saturation analyses. Based on the American Academy of Sleep Medicine (AASM), PSG is typically split into segments of 30 s and classified into five stages: wakefulness (W), non-rapid eye movement sleep (N1, N2, and N3), and rapid eye movement (REM) sleep [9].

There are two primary methods for sleep staging based on PSG: the manual sleep staging method and the automatic sleep staging method. The manual sleep staging method relies on the experience of clinical experts, leading to low accuracy and efficiency [10].

The automatic sleep staging method automatically classifies different stages by using machine learning and deep learning. Machine learning methods classify sleep stages based on manually selected signal features and machine learning models. Methods such as Support Vector Machine (SVM) [11], Random Forest (RF) [12,13], and Decision Tree (DT) [14] achieve high classification performance (e.g., accuracy, precision, and F1-score). However, for high-dimensional PSG data, machine learning methods depend on manual labeling, which leads to inefficiency and sleep information loss during the analysis process [15].

In contrast, deep learning methods achieve automatic sleep staging by constructing multi-layer neural networks. Convolutional Neural Networks (CNNs) [16], Recurrent Neural Networks (RNNs) [17,18], and Attention Mechanisms [19] are methods that automatically extract features from PSG. Convolutional neural networks (CNNs) learn the time–frequency features of PSG through multiple layers of convolution and pooling. These operations map the features to sleep stages, demonstrating good regional feature capture capabilities [20,21]. For instance, Arnaud et al. [16] designed a one-dimensional CNN model specifically for processing one-dimensional signals in sleep staging. This model achieved favorable results. However, CNNs struggle with multi-channel sleep staging data. The fixed convolutional kernel of CNNs leads to the capture of time–frequency characteristics and the parameter volume to surge in deep layers [22].

Recurrent Neural Networks (RNNs) leverage recurrent structures and memory units to process time series data, capture temporal dependencies, and classify sleep stages [23]. However, RNNs struggle with capturing long-term dependencies in sleep data. This limitation motivated the development of Long Short-Term Memory (LSTM) and Gated Recurrent Unit (GRU), which incorporate gating mechanisms to control information flow and effectively map these connections. Supratak et al. [18] proposed DeepSleepNet, which extracts time–frequency features through two CNN branches and employs bidirectional LSTM to learn sleep stage transitions. The model achieved an overall accuracy of 82%, a 3% improvement over traditional manual feature extraction methods. Both LSTM and GRU handle long sequences. GRU simplifies the structure using an update gate while maintaining performance comparable to LSTM. However, the sequential nature of LSTM structures hinders parallel execution [24].

Attention mechanisms assign differential weights to input features, enabling the model to concentrate on the ones most pertinent for classification. Attention mechanisms can handle long sequence data efficiently [19]. Dai et al. [19] proposed MultiChannelSleepNet. This model is based on the Transformer architecture, which integrates positional encoding to maintain the temporal link between features. Additionally, this model achieves good results in extracting single-channel features and fusing multi-channel features. However, attention mechanisms increase model complexity and computational costs and require training data [25].

In summary, the deep learning-based multi-channel sleep staging methods have achieved notable accomplishments thus far. But two major challenges remain: first, the ability to effectively extract salient features from multi-channel sleep data remains limited. Deep learning methods mainly use single- or dual-channel data, ignoring spatial correlations in multi-channel EEG. They employ fixed-size convolution kernels, which restricts the extraction of contextual information from time–frequency images. Second, excessive model parameters hinder efficiency improvements. As sleep models increase in complexity, numerous deep learning models experience a significant surge in parameter volume. This leads to high computational costs, low efficiency, and significant memory usage. Therefore, the research gap lies in the lack of lightweight models that can both (i) effectively capture salient and complementary features from multi-channel PSG data, particularly the spatial correlations among EEG channels, and (ii) reduce computational burden for deployment in resource-constrained environments.

In addition, recent studies have emphasized lightweight architectures for sleep staging. EEGNet [26] employs depthwise separable convolutions to build compact networks suitable for real-time EEG decoding. MobileNet-based approaches [27] extend this principle and are widely used in mobile and embedded scenarios due to their efficiency. More recently, hybrid models such as SalientSleepNet [28] integrate convolutional backbones with attention modules to enhance feature discrimination while keeping parameter counts relatively low. Building upon these advances, our proposed LMCSleepNet differs in three aspects: (1) it integrates continuous wavelet transform to enhance frequency domain representations of multi-channel PSG; (2) it employs multi-scale dilated convolutions to capture contextual dependencies across different time–frequency resolutions; and (3) it introduces CBAM attention on top of a lightweight ResNet18 backbone with depthwise separable convolutions. These innovations enable LMCSleepNet to achieve a better balance between accuracy and computational efficiency, particularly in multi-channel scenarios.

To address these challenges, this work proposes a lightweight multi-channel sleep staging network (LMCSleepNet). This model includes four modules. The first module applies continuous wavelet transform to convert EEG-Fpz-Cz, EEG-Pz-Oz, and EOG signals into time–frequency images. This step enhances the frequency domain features. The second module uses a multi-scale convolution module to extract time–frequency features at various scales from the three signals. This process enriches both key features and contextual information. The third module employs an improved lightweight ResNet18 model to extract and fuse time–frequency features across different channels. Standard convolutions are replaced with depthwise separable convolutions to reduce the parameter of model count and improve computational efficiency. The final module integrates the Convolutional Block Attention Module (CBAM) to enhance the spatial correlation between channels, which increases the accuracy of the classification.

Unlike generic lightweight architectures such as MobileNet, which is designed primarily for image recognition, or EEGNet, which focuses on single-channel EEG processing, LMCSleepNet is specifically optimized for multi-channel sleep staging. Its novelty lies in the integrated use of CWT for frequency–domain enhancement, multi-scale dilated convolutions for contextual feature extraction, depthwise separable convolutions for lightweight efficiency, and CBAM for inter-channel correlation modeling. To the best of our knowledge, this particular combination has not been reported in previous lightweight sleep staging models, making LMCSleepNet distinctive in both architectural design and application.

## 2. Design Based on LMCSleepNet Model

To address the challenges of feature extraction inefficiency and high parameter counts in multi-channel sleep staging (EEG-Fpz-Cz, EEG-Pz-Oz, and EOG), this work proposes a lightweight automatic sleep staging network based on continuous wavelet transform and multi-scale convolution, as shown in Figure 1. The model consists of four modules: a data preprocessing module, a feature extraction module, a feature fusion module, and a classification module. The first module involves data preprocessing module, which leverages continuous wavelet transform to convert one-dimensional time series signals into time–frequency images. This enhances the frequency domain features. The second module is the feature extraction module, which utilizes convolution kernels with varying dilation rates for multi-scale feature extraction. It replaces standard convolutions with depthwise separable convolutions. This decreases the parameter volume of the model and enhances computational efficiency. The third module is the feature fusion module, which integrates features from different channels through the CBAM to enhance the spatial correlations among the multi-channels. The fourth module is the classification module, which outputs the extracted features to a fully connected layer. This fully connected layer ultimately generates sleep stage results through a softmax layer.

### 2.1. Data Preprocessing Module

The continuous wavelet transform (CWT) is a multi-scale time–frequency analysis method for PSG, capable of extracting features at various scales [29]. This method analyzes complex non-stationary time domain signals. It achieves this by stretching and translating the mother wavelet function. Smaller time windows are used at high frequencies, while larger ones are used at low frequencies. To further improve efficiency and robustness, median downsampling is employed, which preserves critical features while reducing noise and computational complexity. In this work, CWT, combined with median downsampling, is employed for data preprocessing, which enhances the frequency domain features of PSG and the processing efficiency of model, as shown in Figure 2. Unlike conventional approaches that directly use full-resolution spectrograms or CWT results, LMCSleepNet employs CWT combined with median downsampling to reduce redundant temporal points while preserving essential frequency domain features. This design choice improves computational efficiency and makes the model more suitable for lightweight deployment.

CWT transforms one-dimensional time series PSG into time–frequency images for model input, enhancing the frequency domain features of the signals. The computation involves the mother wavelet function, scale parameter, translation parameter, and integration operation. The mother wavelet function governs the resolution and analytical capability of the time–frequency image. The Morlet wavelet, with its Gaussian envelope and complex exponential oscillations, is particularly effective at analyzing signals with continuous frequency components. The calculation of the Morlet wavelet is as follows:(1)$$\phi (t)=e^{-\frac{t^{2}}{2}}e^{i\omega_{0}t}$$

In (1), $e^{-\frac{t^{2}}{2}}$ represents the Gaussian function, which determines the envelope shape of the wavelet; $e^{i\omega_{0}t}$ is the complex exponential function; and $\omega_{0}$ is the central frequency, which determines the oscillation frequency of the wavelet.

The CWT is calculated as follows:(2)$$CWT(a;\;\tau )=\frac{1}{\sqrt{a}}\int_{-\infty }^{\infty }f(t)\psi (\frac{t-\tau }{a})dt$$

In (2), f(t) represents the non-stationary PSG; a is the scale parameter; and τ is the translation parameter. When the Morlet wavelet undergoes scaling (a change in scale parameter a), the effects are as follows: when a > 1, the wavelet becomes wider in the time domain and narrower in the frequency domain, primarily used for analyzing low-frequency signals; when 0 < a < 1, the wavelet becomes narrower in the time domain and wider in the frequency domain, primarily used for analyzing high-frequency signals. This work sets the scale parameter a = (1, 32) to extract signal features at different frequencies. The scale range (1, 32) is designed to match the characteristics of EEG signals and meet the needs of effective time–frequency analysis. It covers key EEG frequency bands, including delta (0.5–4 Hz), theta (4–8 Hz), alpha (8–12 Hz), beta (12–30 Hz), and higher frequencies. The translation (change in translation parameter τ) of the Morlet wavelet allows the wavelet to shift along the time axis. This adjustment enables the analysis of signal characteristics at different time points.

The PSG data are divided into multiple continuous segments of 30 s, forming a sequence as follows:(3)$$x_{seg}^{\left(i\right)}=x[i:i+30f_{s}],\;i\in [0,\;1,\;\cdots ,\;n_{seg}]$$

Using wavelet transform, the wavelet time–frequency maps are calculated for each 30 s segment of the raw PSG. The PSG data include EEG-Fpz, EEG-Pz, and EOG.(4)$$x_{cwt}^{\left(i\right)}=CWT(x_{seg}^{\left(i\right)})$$

The wavelet time–frequency maps of the continuous segments are reassembled into a sequence, as follows:(5)$$x_{cwt}=[x_{cwt}^{\left(0\right)},\;x_{cwt}^{\left(1\right)},\;\cdots ,\;x_{cwt}^{\left(n_{seg}\right)}]$$

In (3)–(5), $f_{s}$ is the sampling frequency; x is the raw PSG; $x_{seg}^{\left(i\right)}$ is the i-th segment after segmentation; $n_{seg}=\lfloor L_{raw}/30f_{s}\rfloor$ is the total number of 30 s segments obtained from the raw PSG; $L_{raw}$ is the length of the raw PSG; ⌊•⌋ denotes the floor function; $x_{cwt}^{\left(i\right)}$ is the wavelet time–frequency map obtained from the i-th segment after wavelet transform; and $x_{cwt}$ is the sequence composed of the wavelet time–frequency maps of multiple continuous segments.

CWT has the ability to analyze signals at multiple scales. This method needs to calculate wavelet coefficients at different scales and positions for every time point. This method raises the computational load of the model [30].

To reduce data dimensions and improve neural network training speed—for data dimension reduction and neural network training speed enhancement—methods such as average downsampling [31], max downsampling [32], and median downsampling [33] are used. Given the non-stationary and complex characteristics of PSG, this work employs median downsampling to reduce the number of time points in the time–frequency images to 200. Median downsampling calculates the median of sample values within each time window. This method reduces noise and outliers while preserving feature information. It also improves the efficiency of model processing [33], as shown in Figure 2.

The wavelet time–frequency map is compressed by calculating the median of values within a window for time sampling points. The median value of each window is used as the compressed value, which reduces the temporal dimension of the time–frequency map. This method enhances the training efficiency of the model while preserving the essential signal features as follows:(6)$$x_{zcwt}(t)=median(x_{cwt}(\tau )),\;\tau \in [t,\;t+m]$$

In (6), m represents the number of continuously compressed points; median(⋅) denotes the median value of the m points within the window; τ refers to the time points within [t, t+m] the time interval; and $x_{zcwt}$ is the compressed wavelet time–frequency map.

To determine the impact of the size of time sampling points obtained from the continuous wavelet transform on model performance, comparative experiments with different time sampling point counts were conducted in this work. The original time–frequency map had a sampling duration of 30 s and a sampling frequency of 100 Hz, resulting in a time sampling point count of 3000 (30 s × 100 Hz). The data, originally consisting of 3000 time-sampling points, was compressed using median calculation within a window to achieve reductions to 1/30, 1/20, 1/15, 1/12, and 1/10 of the original size, yielding new time sampling point counts of 100, 150, 200, 250, and 300. The experimental results are shown in Figure 3.

From the experimental results in Figure 3a, the Cohen’s kappa (κ) value of the model is relatively low when the number of time sampling points falls below 200. The reason is that fewer time sampling points reduce the time resolution of time–frequency images, making it difficult to capture the complex dynamic changes during sleep.

When the number of time sampling points exceeds 200, there is no significant enhancement in accuracy. The reason is that increasing the number of time sampling points enhances the time resolution of time–frequency images. However, it also introduces redundant information, which increases the computational cost of model. This redundancy can lead to overfitting and negatively affect the accuracy of sleep classification. Figure 3b–d shows time–frequency patterns with different time sampling points. The results indicate that when the number of time sampling points is below 200, there are fewer features, whereas above 200, there is a greater abundance of feature information, which may lead to redundancy [33].

### 2.2. Feature Extraction Module

#### 2.2.1. Mult-Scale Dilated Convolution Module

Multi-scale dilated convolution (MSDC) uses multiple dilated convolutions to expand the receptive field by adding gaps in the convolution kernels. This method increases the receptive field size without increasing the number of parameters, allowing the extraction of global features. Convolutions with different dilation rates capture contextual information at various scales. In sleep staging tasks, time–frequency features often vary in scale across regions of the time–frequency images. A single-scale receptive field may fail to capture these features comprehensively, reducing sleep classification accuracy [34]. To address this, this work designs the MSDC module to extract multi-scale contextual information, as shown in Figure 4a.

In LMCSleepNet, the MSDC module is placed before the ResNet18 backbone to enhance contextual representation of time–frequency maps. This integration is specifically designed to address the multi-scale variability of sleep patterns, and our comparative experiments (Figure 4b) demonstrate that this design significantly improves classification performance compared to single-scale convolutions.

The MSDC module employs three parallel 3 × 3 convolutional kernels with dilation rates of d (where d = 1, 2, 3). By varying k and l of the convolutional kernels, it extracts features from different locations. The variation in dilation rates allows the convolutional kernels to expand and contract, altering the size of the receptive field. When d = 1, the convolutional kernel functions as a standard convolution. When d > 1, the receptive field of the convolutional kernel increases without changing the number of parameters, as shown below:(7)$$y_{d}(i,\;j)=\sum_{k}\sum_{l}x_{zcwt}(i+d\cdot k,\;j+d\cdot l)\cdot w_{d}(k,\;l)$$

In (7), $x_{zcwt}$ represents the compressed wavelet time–frequency image; $w_{d}$ denotes the convolutional kernel, where d indicates different dilation rates; and $y_{d}(i,\;j)$ is the feature map output after convolution.

To preserve multi-scale feature information as much as possible, the different channel features extracted with varying dilation rates d are fused in this work, resulting in the fused features being expressed as follows:(8)$$Y(i,\;j)=Concat(y_{1}(i,\;j),\;y_{2}(i,\;j),\;y_{3}(i,\;j))$$

In (8), Concat represents the concatenation of the feature maps from three different scales along the channel dimension, and Y(i, j) denotes the resulting multi-scale feature map after concatenation.

To determine the impact of the fusion method for the three time–frequency feature maps captured by multi-scale dilated convolution on model performance, both the addition and concatenation of feature maps were tested on the SleepEDF-20 dataset. The input PSG data comprises EEG-Fpz-Cz, EEG-Pz-Oz, and EOG, with a feature map size of 200 × 32 for each channel. The 32 scales in the Morlet wavelet transform represent the analysis of the signal across 32 different frequency bands, while the 200 time points define the time resolution, as shown in Figure 4b.

From the experimental results, using feature concatenation as the fusion method retained more multi-scale features compared to the addition method, resulting in increases of 4.1% in accuracy and 4.7% in the MF1 score. In contrast, the feature addition fusion may lead to the loss of some feature information, weakening the effectiveness of information at different scales. The strategy of combining feature concatenation can better preserve the integrity of multi-channel features, allowing for more accurate classification in the complex task of sleep stage classification.

#### 2.2.2. ResNet18 Module Based on Depth Separability

The residual neural network addresses the problem of vanishing gradients and performance degradation in deep networks by directly mapping the outputs of shallow convolutions to deeper layers [35]. However, as sleep signal data grows rapidly, the computational load for model training has increased significantly. As a result, researchers have focused more on lightweight model designs. Although ResNet18 is already a lightweight model, this work further reduces its parameter count and enhances computational efficiency by replacing standard convolutions with depthwise separable convolutions. Additionally, to prevent overfitting and improve generalization, a dropout method has been added to the network, as shown in Figure 5.

In contrast to conventional ResNet-based models, LMCSleepNet replaces all standard convolutions in ResNet18 with depthwise separable convolutions. This systematic replacement reduces the parameter count from 11.69 M to only 1.49 M, while the residual connections mitigate potential losses in cross-channel feature representation. This adaptation ensures that LMCSleepNet remains lightweight without compromising classification accuracy.

In this work, standard convolutions in the ResNet18 module are replaced with depthwise separable convolutions, which consist of depthwise and pointwise convolutions. Depthwise convolutions process each input channel independently, effectively extracting spatial features while significantly reducing parameters and computational complexity. Pointwise convolutions use 1 × 1 convolutions to linearly combine the outputs of the depthwise convolutions, integrating information from different channels. This approach improves computational efficiency and preserves key feature representations, making it effective for handling complex input data [27]. The formula for depthwise separable convolutions is as follows:(9)$$F_{DSC}(X,\;\{W_{d},\;W_{p}\})=Y_{p}(Y_{d}(X))$$

In (9), $Y_{d}(X)$ represents the feature map obtained from the depthwise convolution operation on the input feature map X; $W_{d}$ and $W_{p}$ are the weights for the depthwise convolution and pointwise convolution; $Y_{p}(\cdot )$ is the output feature map obtained from applying the pointwise convolution to the depthwise convolution output feature map. $F_{DSC}(X,\;\{W_{d},\;W_{p}\})$ denotes the output feature map based on depthwise separable convolutions.

However, while depthwise separable convolutions reduce the number of parameters, they may struggle to capture high-order relationships across channels, which can affect the extraction of complex features. The residual module of ResNet18 performs an identity mapping on the input, which is then directly added to the output of the module. This residual connection mitigates the deficiencies of depthwise separable convolutions in extracting complex features. It enhances the expressive power of the model. The equation is as follows:(10)$$Y=F_{DSC}(X,\;\{W_{d},\;W_{p}\})+X$$

In (10), X represents the input feature map; $F_{DSC}(X,\;\{W_{d},\;W_{p}\})$ is the output feature map after applying depthwise separable convolutions; and Y denotes the feature map after the residual connection.

### 2.3. Feature Fusion Module

The Convolutional Block Attention Module (CBAM) combines channel and spatial attention mechanisms to improve the network’s ability to focus on important information in feature maps, enhancing the feature modeling capability across different positions and channels [36]. With its simple structure, the CBAM requires fewer parameters and computational resources than other attention mechanisms, making it easy to integrate into various networks. In sleep staging, the influence of different signal channels on sleep stages varies, and the correlation among channels is weak. This limits the ability to fully utilize their complementary information, leading to suboptimal classification accuracy. To address this, this work introduces the CBAM to focus on relevant information in time–frequency images and strengthen the correlation among channels, improving the model’s ability to extract important features.

The CBAM consists of two parts: channel attention and spatial attention. The channel attention module compresses the spatial dimensions of the feature map using global average pooling and global max pooling to extract the importance of the target object. The results of these two pooling operations are then concatenated. Channel attention weights are calculated using a fully connected layer and an activation function (e.g., ReLU). These weights are applied to the feature maps of each channel. The channel attention module is formulated as follows:(11)CA(X)=X⊙(σ(FC(AvgPool(X))+FC(MaxPool(X))))(12)FC(•)=Conv(m, α, 1)

In (11) and (12), $X\in R^{H\times W\times C}$ represents the input time–frequency feature map; H is the height of the time–frequency map; W is the width of the time–frequency map; C is the number of channels in the sleep signals. AvgPool and MaxPool denote the computed average and maximum values of the channels; σ is the activation function used to calculate the channel attention weights; and CA(X) represents the weighted feature map obtained by applying the weights to each channel.

The spatial attention module applies average pooling and max pooling to the output of the channel attention module. It then calculates the spatial attention weights using a convolutional layer and an activation function (e.g., Sigmoid). These weights are applied to each pixel of the feature map. Finally, the input feature map is combined with the spatial attention weights to generate the output feature map. The spatial attention module can be represented as follows:(13)SA(X)=X⊙(σ(Conv(concat(AvgPool(X), MaxPool(X)), 1, k)))

In (13), σ represents the Sigmoid function used to calculate the spatial attention weights; SA(X) denotes the application of the weights to each pixel; and k indicates the size of the convolutional kernel.

The CBAM performs an element-wise multiplication of the outputs from the channel attention mechanism and the spatial attention mechanism to obtain the final feature map. The computation of the CBAM is as follows:(14)CBAM(X)=CA(X)⊙SA(X)

In (14), ⊙ represents the element-wise multiplication operation, CBAM(X), resulting in a feature map that achieves global attention and weighting, thereby enhancing the representational capability of the feature map.

Unlike prior studies that mainly apply the CBAM for visual feature refinement, LMCSleepNet integrates the CBAM as a multi-channel fusion mechanism. By jointly modeling inter-channel (EEG–EOG) correlations, the CBAM enhances complementary feature interactions across modalities, thereby improving sleep stage classification beyond simple concatenation strategies.

### 2.4. Classification Module

The classification module constitutes the final stage of LMCSleepNet. After feature extraction and fusion by the preceding modules, the resulting feature maps are flattened and passed into a fully connected (FC) layer. This layer projects the high-dimensional fused features into a lower-dimensional vector corresponding to the number of sleep stages. Finally, the softmax activation function is applied to normalize the output into a probability distribution across the five sleep stages (W, N1, N2, N3, and REM). The predicted class is assigned to the stage with the highest probability.

Mathematically, the classification process can be expressed as follows:(15)$$z=W_{fc}\cdot X_{fused}+b_{fc}$$(16)$$P_{i}=\frac{e^{z_{i}}}{\sum_{j=1}^{C}e^{z_{j}}},\;i=1,\;2,\;\ldots ,\;Cz_{i}i$$
where $X_{fused}$ denotes the fused feature vector obtained from the CBAM, $W_{fc}$ and $b_{fc}$ are the trainable parameters of the fully connected layer, $z_{i}$ is the logit corresponding to the i-th class, C = 5 is the number of sleep stages, and $P_{i}$ represents the predicted probability for the i-th sleep stage.

This module enables LMCSleepNet to map the integrated multi-scale and multi-channel features into discrete sleep stage categories, completing the automatic sleep staging process.

### 2.5. Efficiency Analysis of Modules

To further clarify the contribution of each design choice to efficiency, we provide the following analysis: The wavelet transform produces high-dimensional time–frequency images (3000 sampling points for each 30 s segment). Directly using these images would result in high computational cost. By applying median downsampling, the number of sampling points is reduced to 200 while preserving salient signal characteristics and suppressing noise. This operation lowers data dimensionality, accelerates training, and improves robustness, thereby enhancing efficiency. MSDC enlarges the receptive field through dilation without introducing additional parameters. Compared with stacking extra convolutional layers, MSDC efficiently captures contextual information across multiple temporal scales with lower computational overhead. Thus, it achieves more comprehensive feature extraction without sacrificing efficiency. Standard convolutions are replaced by depthwise separable convolutions, which decompose a convolution into depthwise filtering and pointwise channel mixing. This reduces the number of parameters and floating-point operations by an order of magnitude compared with conventional convolutions. As a result, memory usage and computation time are significantly reduced, while the representational ability of ResNet18 is preserved. Instead of deepening or widening the network, CBAM improves feature utilization by adaptively emphasizing informative channels and spatial regions. This lightweight attention mechanism only adds negligible parameter overhead but enhances the discriminative power of the extracted features. Consequently, the CBAM achieves a favorable accuracy–efficiency trade-off.

To summarize, the four modules in LMCSleepNet interact in a complementary manner to address the two major challenges of sleep staging. First, the CWT module transforms raw PSG signals into informative time–frequency representations, providing a rich basis for downstream feature learning. The MSDC module then extracts multi-scale contextual features, which alleviates the limitation of fixed convolution kernels and improves feature extraction efficiency. Next, the ResNet18 backbone equipped with depthwise separable convolutions encodes these features with greatly reduced parameter complexity, ensuring lightweight model design. Finally, the CBAM enhances inter-channel correlations and emphasizes salient spatial–temporal features without introducing significant computational burden. Through this coordinated design, LMCSleepNet achieves both effective feature extraction and high parameter efficiency, thereby directly overcoming the two challenges identified in the Introduction.

## 3. Experiment and Result Analysis

### 3.1. Dataset

To evaluate the proposed LMCSleepNet, this work uses two publicly available datasets: SleepEDF-20 and SleepEDF-78, as shown in Table 1.

SleepEDF-20 is a subset of the Sleep-EDF Extension Dataset (2013 version) [37,38], which includes studies from Sleep Cassette (SC) and Sleep Telemetry (ST). The SC study involved 20 subjects aged from 25 to 34 and investigated the relationship between age and sleep in healthy populations. The ST study focused on the effects of diazepam on sleep. To avoid the influence of other factors (such as medications) on the experimental results [39,40], this work uses the SC subset as the experimental dataset. Each PSG recording contains two EEG channels (Fpz-Cz, Pz-Oz), one horizontal EOG channel, and one EMG channel, with all EEG and EOG signals sampled at 100 Hz. Sleep experts manually classified these recordings into eight categories (W, N1, N2, N3, N4, MOVEMENT, and UNKNOWN) based on the R&K standard. In the experiments, the AASM standard was used to combine the N3 and N4 stages into one sleep stage N3 [41], while the MOVEMENT and UNKNOWN stages were removed. This experiment evaluates the proposed model using the EEG-Fpz-Cz, EEG-Pz-Oz, and EOG channels, with a sampling rate of 100 Hz.

SleepEDF-78 is a subset of the Sleep-EDF Extension Dataset (2018 version) [37,38], which includes 78 subjects aged from 25 to 101, with 153 whole-night PSG sleep recordings. Similarly to SleepEDF-20, each subject underwent two PSG recordings. The manual scoring criteria and 30 s PSG stage labels are the same as those in SleepEDF-20. This experiment evaluates the proposed model using the EEG-Fpz-Cz, EEG-Pz-Oz, and ROC-LOC EOG (horizontal) channels, with a sampling rate of 100 Hz.

### 3.2. Experimental Setup and Model Parameters

This work focuses on training and evaluating a lightweight model for automatic sleep staging. The model is based on a multi-scale ResNet18 network, which uses depthwise separable convolutions to improve efficiency. Table 2 describes the structure of the LMCSleepNet model and its corresponding parameters. The input image size of 200 × 32 represents 200 sampled time points along the time axis and 32 frequency scales analyzed by the wavelet transform.

The experiments used Python 3.9 and the PyTorch 1.9.0 deep learning framework, with CUDA 11.1 for GPU acceleration. The proposed model was implemented on an RTX 4090D GPU with 24 GB memory. The optimizer selected for this experiment was Adam [42], $\beta_{1}=0.9$, $\beta_{2}=0.999$, with a learning rate of $5\times 10^{-6}$. During training, a batch size of 64 was used. This work introduced an early stopping strategy, halting the training process if the accuracy on the validation dataset did not improve after 20 epochs.

To evaluate model performance, this work employed 10-fold cross-validation, dividing the dataset into 10 parts. For each training iteration, 9 parts were used as the training set and the remaining 1 part as the validation set. The batch size for the training set was 64, and the training time for each fold of cross-validation was approximately 0.8 h. The model was trained using a weighted cross-entropy loss function. To prevent overfitting, L2 regularization was applied to the loss function, with a weight of $10^{-3}$.

To ensure stable and efficient training, the hyperparameters of LMCSleepNet were carefully selected based on prior studies and validation experiments. The Adam optimizer with an initial learning rate of 5 × 10^−6^ was chosen because of its adaptive learning ability and proven effectiveness in EEG and PSG classification tasks [16,17]. The batch size was set to 64, consistent with earlier work on biomedical signal classification [39], balancing computational efficiency and convergence stability. For weight decay, an L2 regularization coefficient of 1 × 10^−3^ was employed, following best practices in lightweight ResNet-based models [34]. An early stopping strategy with a patience of 20 epochs was adopted to prevent overfitting, consistent with recent recommendations [41].

Moreover, we validated the number of time sampling points in the wavelet time–frequency maps through comparative experiments (100, 150, 200, 250, and 300 points). As shown in Figure 3, 200 sampling points achieved the best balance between classification accuracy and computational cost, justifying this choice. These experimental results further confirm the robustness of the selected hyperparameters.

To address the class imbalance issue in the SleepEDF datasets, we employed a weighted cross-entropy loss function, where the class weights were set inversely proportional to the number of samples in each sleep stage. This strategy encourages the model to focus more on minority classes such as N1 and alleviates the negative impact of the imbalanced data distribution. In preliminary experiments, we also evaluated focal loss, but it resulted in less stable convergence compared to weighted cross-entropy. Therefore, all the experimental results reported in this work are based on the weighted cross-entropy loss function.

In addition, several strategies were employed to improve the robustness and generalization of the model. Specifically, median downsampling was adopted to reduce the redundancy of wavelet time–frequency maps while being more robust to noise and outliers compared with average or max downsampling. To address the class imbalance problem in the SleepEDF datasets, a weighted cross-entropy loss was employed to increase the penalty for minority classes (e.g., N1) and improve macro-average performance. Furthermore, an early stopping strategy was used to prevent overfitting and enhance generalization by terminating the training process when the validation accuracy did not improve after 20 consecutive epochs.

### 3.3. Evaluation Criteria

This work uses precision (PR), recall (RE), and F1-score (F1) as per-class metrics to evaluate the LMCSleepNet model. Other metrics, including accuracy (ACC), macro-average F1-score (MF1) [43], and Cohen’s kappa (κ) [28], were used to evaluate the overall model performance. The F1 provides a more accurate evaluation of the model’s performance on imbalanced data classification. The MF1 is the average of the F1 across all classes. κ reflects the consistency between the model’s sleep stage results and those of sleep experts. The formulas for PR, RE, F1, ACC, MF1, and κ are as follows:(17)$$PR=\frac{TP}{TP+FP}$$(18)$$RE=\frac{TP}{TP+FN}$$(19)$$F1_{i}=2\cdot \frac{RE\cdot PR}{RE+PR}$$(20)$$ACC=\frac{TP+TN}{TP+FN+TN+FP}$$(21)$$MF1=\frac{\sum_{i=1}^{n}F1_{i}}{n}$$(22)$$\kappa =\frac{\frac{\sum_{i=1}^{n}\chi_{ii}}{N}-\frac{\sum_{i=1}^{n}\left(\sum_{j=1}^{n}\chi_{ij}\sum_{j=1}^{n}\chi_{ji}\right)}{N^{2}}}{1-\frac{\sum_{i=1}^{n}\left(\sum_{j=1}^{n}\chi_{ij}\sum_{j=1}^{n}\chi_{ji}\right)}{N^{2}}}$$

In (15)–(20), TP, TN, FN, and FP represent the counts for each category in the experimental classification results, where TP indicates true positives (correctly classified positive instances), TN indicates true negatives (correctly classified negative instances), FN indicates false negatives (incorrectly classified negative instances), and FP indicates false positives (incorrectly classified positive instances). N represents the total number of 30 s samples in the test set, and n represents the number of classes. In this work, n = 5, x_ii_ (1 ≤ i ≤ 5) is set to the diagonal values of the confusion matrix.

### 3.4. Comparison and Analysis of Experimental Results

The proposed LMCSleepNet model was experimentally evaluated on two public datasets, SleepEDF-20 and SleepEDF-78. Table 3 presents the training results of the model on the small-sample SleepEDF-20 dataset, including the confusion matrix for the five sleep stages and the performance metrics (PR, RE, and F1) for each stage. Figure 6 visualizes the confusion matrix of the sleep stages.

As shown in Table 3, the LMCSleepNet model achieves over 86% PR for the W, N2, N3, and REM stages. Notably, the PR, RE, and F1 scores for the W stage exceed 92%, the highest among all five classes. This demonstrates the model’s ability to effectively extract sleep features for the W stage, because W stage samples dominate the training dataset. However, the PR, RE, and F1 scores for the N1 stage are below 60%, with many N1 samples misclassified as W, N2, or REM stages. This occurs because the N1 stage is a brief transitional phase in the sleep cycle, resulting in fewer samples and fewer learnable features. Additionally, the lambda wave, a key feature of the N1 stage, often appears in N2 and REM stages, making it difficult for the model to distinguish these phases.

Although the overall performance of LMCSleepNet is promising, the classification of the N1 stage remains relatively weak. As shown in Table 3 and Figure 6, many N1 samples are incorrectly classified into W, N2, and REM stages. Specifically, 13.8% of N1 samples are predicted as W, 21.9% as N2, and 14.5% as REM. This misclassification pattern indicates that the model struggles to distinguish the transitional characteristics of N1. The main reason lies in the low-amplitude mixed-frequency (LAMF) activity of N1, which exhibits similarities to alpha waves in W, spindle activity in N2, and theta rhythms in REM. Additionally, the limited number of N1 samples (6.5% in SleepEDF-20 and 11% in SleepEDF-78) exacerbates class imbalance, reducing the discriminability of the model in this stage.

The proposed LMCSleepNet model was experimentally evaluated on two public datasets, SleepEDF-20 and SleepEDF-78. Table 3 presents the training results of the model on the small-sample SleepEDF-20 dataset, including the confusion matrix for the five sleep stages and the per-class performance metrics (PR, RE, and F1). Figure 6 visualizes the confusion matrix of the sleep stages.

As shown in Table 3, LMCSleepNet achieves over 86% PR for the W, N2, N3, and REM stages. Notably, the W stage obtains the highest PR, RE, and F1 scores (all above 92%), indicating strong discriminative ability. However, the N1 stage remains challenging, with PR, RE, and F1 below 60%, due to its transitional nature and overlapping features with neighboring stages.

To further evaluate the performance of LMCSleepNet, we compared it against existing baselines. For clarity, we separated the baselines into two categories: single-channel models and multi-channel models. (1) Comparison with single-channel baselines. The single-channel baselines include DeepSleepNet [17], SleepEEGNet [18], and TinySleepNet [44]. As shown in Table 4, LMCSleepNet achieves an accuracy of 88.2% on SleepEDF-20, improving by 6.2%, 3.9%, and 2.8% over DeepSleepNet, SleepEEGNet, and TinySleepNet, respectively. The macro F1-score also increases by 5.5%, 3.7%, and 1.9%. Moreover, the per-class F1-scores for W, N1, N2, N3, and REM all exceed those of the single-channel baselines. These improvements stem from LMCSleepNet’s ability to integrate multi-channel PSG signals and extract richer contextual information through multi-scale dilated convolutions. Although LMCSleepNet has slightly more parameters than TinySleepNet (1.49 M vs. 1.3 M), it achieves higher accuracy (+1.9%), indicating a favorable trade-off between complexity and performance. (2) Comparison with multi-channel baselines. The multi-channel baselines include MultiChannelSleepNet [19] and SalientSleepNet [45]. LMCSleepNet achieves an accuracy improvement of 1.0% over MultiChannelSleepNet and 0.7% over SalientSleepNet on SleepEDF-20. For the W, N2, and N3 stages, the F1-scores of LMCSleepNet surpass both baselines, demonstrating better utilization of inter-channel correlations. Importantly, the parameter count of LMCSleepNet (1.49 M) is only 11.5% of MultiChannelSleepNet (13 M), confirming its lightweight design. Compared with SalientSleepNet (0.9 M), LMCSleepNet requires 0.59 M more parameters, but this increase is justified by higher classification accuracy (+0.7%) and enhanced channel interaction modeling via the CBAM.

Figure 7a further illustrates the trade-off between accuracy and parameter count. LMCSleepNet achieves a favorable balance, maintaining high accuracy while keeping the model size significantly smaller than most baselines.

On the larger SleepEDF-78 dataset, similar trends are observed (Table 5). Compared with single-channel baselines (DeepSleepNet, SleepEEGNet, TinySleepNet), LMCSleepNet improves accuracy by 5.6%, 1.3%, and 1.0%, respectively, and consistently achieves competitive F1-scores across stages. When compared with multi-channel baselines, LMCSleepNet attains comparable accuracy to SalientSleepNet (+0.0%) and slightly lower than MultiChannelSleepNet (−0.9%), but with far fewer parameters (1.49 M vs. 13 M). Figure 7b confirms that LMCSleepNet offers an effective compromise between accuracy and model complexity, particularly valuable in resource-constrained scenarios.

Although SalientSleepNet employs fewer parameters (0.9 M) compared to LMCSleepNet (1.49 M), the proposed model demonstrates several distinctive advantages. First, LMCSleepNet achieves more consistent improvements across multiple datasets (SleepEDF-20 and SleepEDF-78), highlighting its stronger generalization ability. Second, by incorporating the Multi-Scale Dilated Convolution (MSDC) module, LMCSleepNet captures contextual dependencies at different receptive fields, which enhances the classification of transitional stages such as N1 and REM. Third, the Convolutional Block Attention Module (CBAM) enables more effective integration of complementary information from the EEG and EOG channels, thereby improving the robustness of multi-channel feature fusion. These design choices allow LMCSleepNet to strike a favorable balance between accuracy, robustness, and parameter size. Despite having 0.59 M more parameters than SalientSleepNet, LMCSleepNet remains a lightweight architecture that provides superior stability across datasets and sleep stages.

Additionally, the LMCSleepNet model was evaluated on the larger SleepEDF-78 dataset, as shown in Table 5. The results are derived from the SleepEDF-78 dataset and were obtained through 10-fold cross-validation. An “-“ indicates that the performance metric was not reported in the original literature.

In addition to the performance comparison, it is important to analyze the limitations of the baseline models. DeepSleepNet, despite achieving favorable accuracy, contains 21 M parameters, leading to excessive computational cost and memory usage, which hinders deployment on resource-constrained devices. MultiChannelSleepNet, built on the Transformer architecture, effectively captures cross-channel dependencies but requires 13 M parameters, making it computationally expensive. SleepEEGNet and TinySleepNet are relatively lightweight; however, their reliance on single-channel EEG restricts the integration of complementary multimodal features, resulting in limited generalization. SalientSleepNet is lightweight (0.9 M parameters) but its classification accuracy is still lower than LMCSleepNet, which indicates that reducing parameter count alone may lead to the loss of discriminative features.

By contrast, LMCSleepNet achieves a balance between efficiency and accuracy. Through the use of multi-scale dilated convolutions, depthwise separable convolutions, and CBAM-based feature fusion, LMCSleepNet reduces the parameter count to 1.49 M while maintaining higher accuracy than the majority of baselines. This highlights that the proposed model not only improves feature extraction efficiency but also effectively integrates complementary information across multiple PSG channels, making it more suitable for real-world and resource-limited scenarios.

We further compared the training efficiency of LMCSleepNet with other baseline models. The comparison of single-sample training time on SleepEDF-20 and SleepEDF-78 is presented in the Appendix A. The results show that LMCSleepNet achieves comparable efficiency to other lightweight models such as TinySleepNet and SalientSleepNet, while being substantially faster than large-scale models including DeepSleepNet and MultiChannelSleepNet. This confirms that LMCSleepNet achieves a favorable balance between accuracy, parameter size, and computational efficiency.

For single-channel baseline models like DeepSleepNet, SleepEEGNet, and TinySleepNet, the proposed LMCSleepNet model achieved an accuracy of 84.1%, improvements of 5.6%, 1.3%, and 1.0%. The F1 for the W, N2, and REM stages also exceed those of the baseline models. This improvement is mainly due to the model’s ability to extract important wave features from different channels, which enhances its classification accuracy.

For multi-channel baseline models like MultiChannelSleepNet and SalientSleepNet, the accuracy of LMCSleepNet model is 0.9% lower than that of MultiChannelSleepNet, but its parameter count is only 11.5% of MultiChannelSleepNet. This is mainly due to LMCSleepNet’s lack of long-term dependencies in large-sample data and the use of depthwise separable convolutions instead of standard convolutions. As a result, its feature extraction capability is limited, which affects classification accuracy. Additionally, although SalientSleepNet has the smallest parameter count (0.9 M), its F1 for the W stage is still 0.9% lower than that of LMCSleepNet. This suggests that LMCSleepNet model is more effective at extracting and fusing multi-scale features.

Figure 7b shows the relationship between parameter count and accuracy for LMCSleepNet model and the baseline models. The results demonstrate that, although LMCSleepNet model has slightly lower accuracy, it strikes a good balance between accuracy and model complexity on the larger dataset. It maintains a lightweight structure while offering more robust performance in sleep stage classification tasks.

In addition to accuracy and parameter comparisons, we further evaluated the inference latency of LMCSleepNet to assess its real-time applicability on different hardware platforms. On a workstation equipped with an RTX 4090D GPU, the average inference time for a 30 s PSG segment (input size 200 × 32) was approximately 4.8 ms, which easily meets the requirement for online sleep stage classification. To examine performance on resource-constrained devices, LMCSleepNet was deployed on a NVIDIA Jetson Nano (ARM Cortex-A57 CPU, Maxwell GPU, 4 GB RAM). The model achieved an average inference latency of 28.6 ms per 30 s segment, which is still substantially below the 30 s decision window typically required in clinical and wearable sleep monitoring scenarios. These results demonstrate that the lightweight design of LMCSleepNet not only reduces the parameter count and memory usage but also enables efficient real-time inference across both high-performance and embedded devices. This confirms the practical trade-off between lightweight structure and latency, showing that LMCSleepNet maintains robustness for real-world deployment.

To further normalize the comparison between classification accuracy and model complexity, we introduced the efficiency metric accuracy-to-parameters ratio (Acc/M), which measures the accuracy achieved per million parameters. As shown in Table 4 and Table 5, LMCSleepNet obtains an Acc/M of 59.2%/M on the SleepEDF-20 dataset and 56.4%/M on the SleepEDF-78 dataset. These values are significantly higher than those of larger models such as DeepSleepNet (3.9%/M and 3.7%/M) and MultiChannelSleepNet (6.7%/M and 6.5%/M), and are comparable to other lightweight baselines. This confirms that LMCSleepNet strikes a superior balance between classification accuracy and lightweight design, making it more suitable for deployment in resource-constrained environments.

### 3.5. Ablation Experiment

To validate the effectiveness of the lightweight LMCSleepNet model, four groups of ablation experiments were designed based on ResNet18, with each group using the same training parameters. The experimental results are shown in Table 6, where “√” indicates the inclusion of the respective module. The results reveal that when using the basic ResNet18 module, the model achieved an ACC of 87.1%, a κ of 0.82, and an MF1 of 80.7%, with a parameter count of 11.69 M. Adding the CBAM improved the detection ACC by 0.9%, while the κ and MF1 increased by 0.01 and 1.1%. This improvement is mainly due to the CBAM enhancing multi-channel feature fusion and increasing the correlation between channels. To further reduce the model’s parameter count, standard convolutions in the ResNet18 module were replaced with depthwise separable convolutions. While this led to a 0.7% decrease in ACC, the parameter count was reduced to only 12.7% of the base ResNet18 model. This reduction is due to the depthwise separable convolutions, which effectively extract features using depthwise and pointwise convolutions, lowering computational complexity and making the model more lightweight. To balance the lightweight design and accuracy, an MSDC module was added. This module maintained its lightweight nature while increasing ACC to 88.2%, representing a 0.9% improvement. In the classification results, the F1 for the W, N2, and N3 stages exceeded 90%, and the F1 for the REM stage surpassed 85%. For the N1 stage, which had a smaller sample size, the F1 was 2.5–6.6% higher than in the other ablation experiments.

Figure 8a shows the relationship between the parameter count and accuracy of the LMCSleepNet model across different modules. We clearly observe that the LMCSleepNet model achieves a lightweight design while enhancing accuracy. Figure 8b reveals that the proposed LMCSleepNet model outperforms other ablation experiments in terms of accuracy, kappa, and F1-score, validating its effectiveness and superiority in sleep stage classification. Additional ablation experiments on the SleepEDF-78 dataset (see Appendix A) further demonstrate that both the CBAM and MSDC consistently improve the classification performance, confirming their generalization ability across datasets.

Furthermore, to provide intuitive evidence of the effectiveness of the MSDC and CBAM, we visualized their feature representations. As shown in the Appendix A, the MSDC module enables the extraction of multi-scale contextual information, while the CBAM highlights the most discriminative regions through channel and spatial attention. These visualizations confirm that the proposed modules contribute significantly to enhancing feature representation and improving classification performance.

## 4. Conclusions

The work addresses the challenges of feature extraction inefficiency and high parameter counts in multi-channel sleep staging (EEG-Fpz-Cz, EEG-Pz-Oz, and EOG). This work proposes a lightweight automatic sleep staging network based on continuous wavelet transform and multi-scale convolution. The model consists of four components: data preprocessing, a feature extraction module, a feature fusion module, and a classification module. Preprocessing converts one-dimensional signals to time–frequency images using wavelet transform. Feature extraction employs multi-scale kernels with varying dilation rates, and it is implemented based on ResNet18 with depthwise separable convolutions. Feature fusion integrates features via the CBAM to enhance spatial correlations. Classification outputs features to a fully connected layer, resulting in sleep stages via softmax.

The LMCSleepNet model was evaluated on the public datasets SleepEDF-20 and SleepEDF-78, achieving accuracies of 88.2% and 84.1% while reducing the parameter count to 1.49 M. The model also examined the impact of the number of time samples in the wavelet time–frequency images on classification performance (ACC, κ, and MF1). It analyzed the relationship between different time sample counts and model performance. A choice of 200 time samples was made because fewer than 200 resulted in insufficient image resolution, while more than 200 introduced redundant information, which causes overfitting and reduces classification accuracy. Furthermore, the LMCSleepNet model examined the impact of the multi-scale dilated convolution module’s fusion method on classification performance (ACC and MF1). It also analyzed how different feature map fusion methods, such as concatenation or addition, affected model performance. The concatenation method was chosen as it preserves multi-scale feature information better than the addition method.

Despite the overall effectiveness of LMCSleepNet, the classification of the N1 stage remains a challenge due to sample scarcity and waveform similarity with adjacent stages. To improve performance in this transitional stage, future research may consider the following: N1-specific feature enhancement—developing modules sensitive to low-amplitude mixed-frequency activity or vertex sharp waves to strengthen stage-specific representations. Data augmentation strategies—employing advanced techniques such as GAN-based signal synthesis or mixup strategies to increase the diversity and quantity of N1 samples. Imbalance-aware learning—integrating cost-sensitive loss functions (e.g., focal loss or class-balanced loss) to mitigate the effect of class imbalance and emphasize minority stage learning. These strategies are expected to enhance the discriminability of N1 features and further improve the robustness of LMCSleepNet in sleep staging applications.

Furthermore, the LMCSleepNet model showed relatively poor performance on larger sample datasets. This was mainly due to the model’s inability to capture long-term dependencies in large-sample data and the use of depthwise separable convolutions instead of standard convolutions. These factors limited its feature extraction capability and affected classification accuracy. In contrast, the LMCSleepNet model performed better on smaller sample datasets, making it more suitable for resource-limited scenarios.

## Figures and Tables

**Figure 1 sensors-25-06065-f001:**
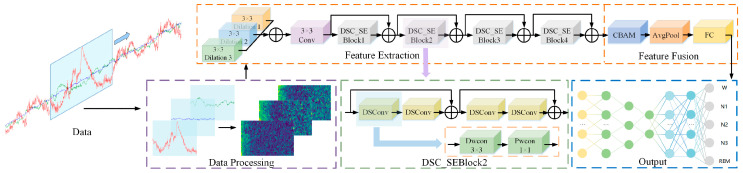
Framework diagram of LMCSleepNet model. Dilation i represents a dilated convolution with a dilation rate of i; DSC_SE Block i denotes the i-th residual block based on depthwise separable convolution; DSConv refers to depthwise separable convolution; Dwcon indicates depthwise convolution; Pwcon stands for pointwise convolution; CBAM represents the channel and spatial attention mechanism; AvgPool refers to the average pooling layer; and FC stands for the fully connected layer.

**Figure 2 sensors-25-06065-f002:**
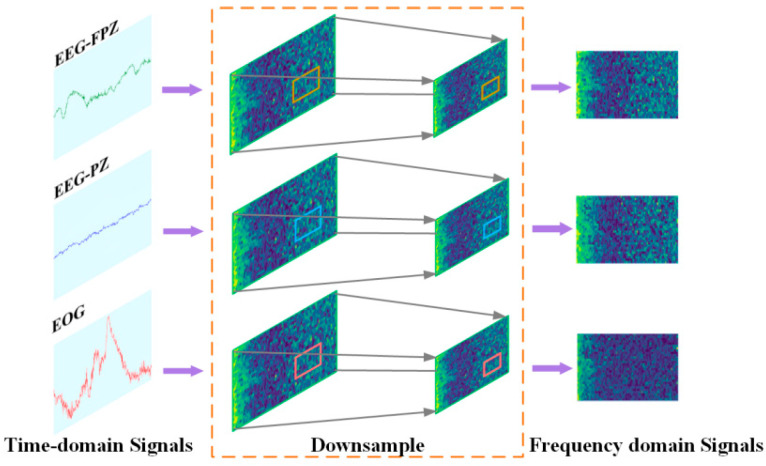
Continuous wavelet transform and data downsampling.

**Figure 3 sensors-25-06065-f003:**
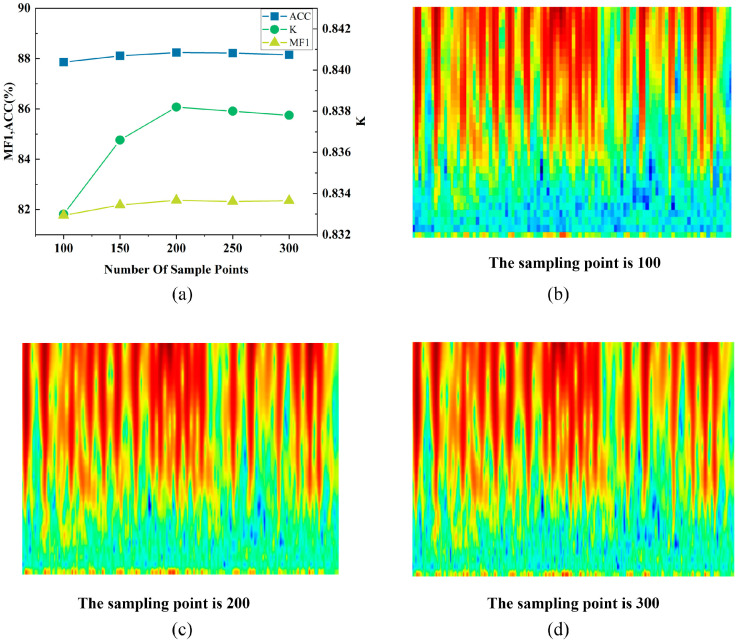
Comparison of the number of sampling points at different times of wavelet transform. (**a**) Performance comparison of sampling points at different times. (**b**–**d**) Time–frequency patterns with different time sampling points.

**Figure 4 sensors-25-06065-f004:**
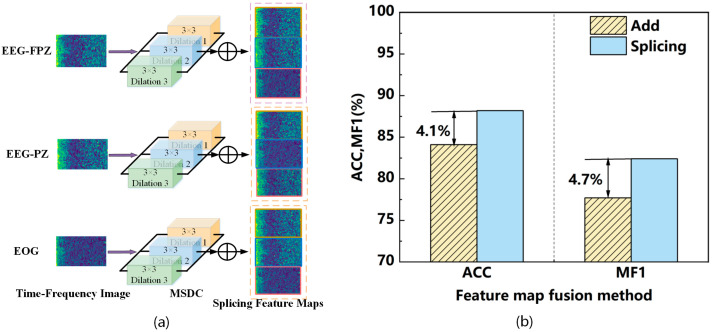
Diagram of the MSDC Model’s Structure and Verification of Its Performance. (**a**) Detailed design diagram of the MSDC model. (**b**) The results of different fusion methods for MSDC features.

**Figure 5 sensors-25-06065-f005:**
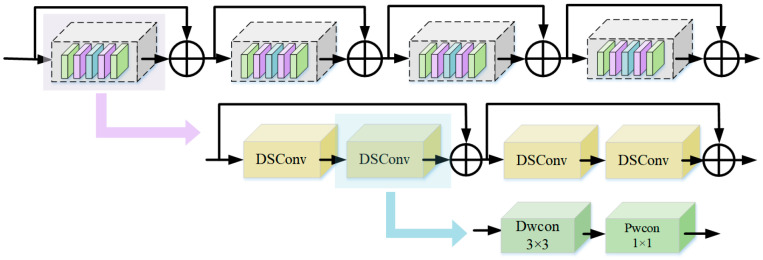
Resnet18 module based on depth separability.

**Figure 6 sensors-25-06065-f006:**
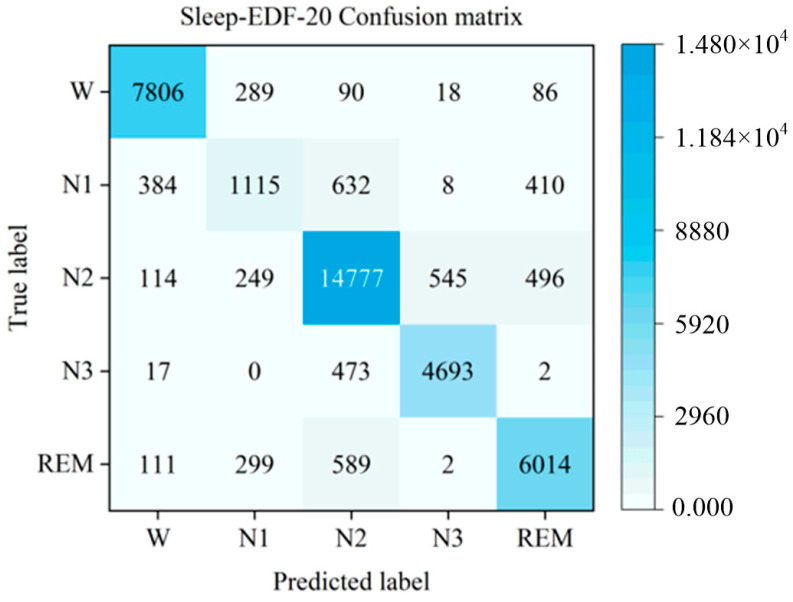
Confusion matrix of LMCSleepNet model on SleepEDF-20 dataset.

**Figure 7 sensors-25-06065-f007:**
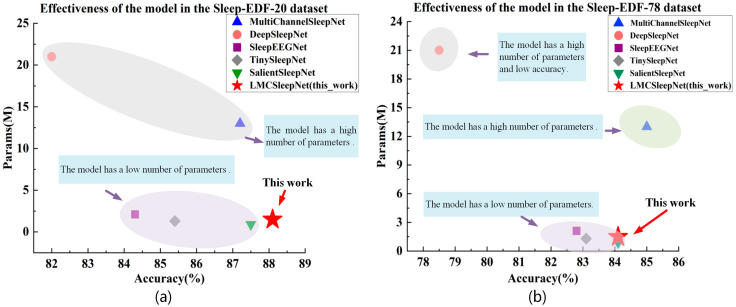
The relationship between model parameter quantity and accuracy on the dataset. (**a**) The relationship between parameter quantity and accuracy of SleepEDF-20 dataset. (**b**) The relationship between parameter quantity and accuracy in the SleepEDF-78 dataset.

**Figure 8 sensors-25-06065-f008:**
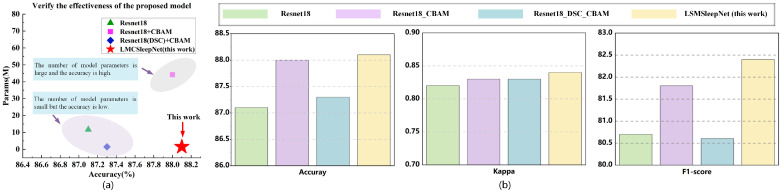
LMCSleepNet model ablation experiment. (**a**) The relationship between the parameter quantity and accuracy of the LMCSleepNet model ablation experiment. (**b**) Comparison of ablation experimental performance of different modules in LMCSleepNet.

**Table 1 sensors-25-06065-t001:** The number and proportion of sleep samples in each stage of the two datasets.

Stage (AASM)	SleepEDF-20 (Number/30 s)	Proportion/%	SleepEDF-78 (Number/30 s)	Proportion/%
W	9118	21.10	66,822	34.00
N1	2804	6.50	21,522	11.00
N2	17,799	41.30	69,132	35.20
N3	5703	13.20	13,039	6.60
REM	7717	17.90	25,835	13.20
Total	43,141	100%	196,350	100

**Table 2 sensors-25-06065-t002:** Structure and parameter settings of the model.

Layer	Kernel Size/ Channels, Stride	Output Size
Input		200×32×3
MSDC	$\left[\begin{array}{l} 3\times 3,\;d=1\\ 3\times 3,\;d=2\\ 3\times 3,\;d=3 \end{array}\right],\;64$	112×112×64
MaxPool	3×3, 64	56×56×64
Layer 1 (DSC)	$\left[\begin{array}{l} 3\times 3,\;64\\ 3\times 3,\;64 \end{array}\right]\times 2$	56×56×64
Layer 2 (DSC)	$\left[\begin{array}{l} 3\times 3,\;128\\ 3\times 3,\;128 \end{array}\right]\times 2$	28×28×128
Layer 3 (DSC)	$\left[\begin{array}{l} 3\times 3,\;256\\ 3\times 3,\;256 \end{array}\right]\times 2$	14×14×256
Layer 4 (DSC)	$\left[\begin{array}{l} 3\times 3,\;512\\ 3\times 3,\;512 \end{array}\right]\times 2$	7×7×512
Channel Attention (CA)	1×1	7×7×512
Spatial Attention (SA)	7×7	7×7×512
Average Pooling	7×7×512	1×1×512

**Table 3 sensors-25-06065-t003:** Training results of the model on the SleepEDF-20 dataset.

True Label	Predicted Label					Performance Metrics		
	W	N1	N2	N3	REM	PR (%)	RE (%)	F1 (%)
W	7844	276	87	14	68	92.7 ± 0.2	94.6 ± 0.2	93.7 ± 0.2
N1	387	1131	614	12	405	59.4 ± 0.3	44.4 ± 0.3	51.5 ± 0.3
N2	109	241	14,858	514	459	89.5 ± 0.3	91.8 ± 0.3	90.6 ± 0.4
N3	17	0	461	4706	1	89.7 ± 0.2	90.8 ± 0.4	90.2 ± 0.2
REM	106	257	582	2	6068	86.7 ± 0.2	86.5 ± 0.2	86.6 ± 0.3

**Table 4 sensors-25-06065-t004:** Comparison of the proposed LMCSleepNet model with other benchmark models on the SleepEDF-20 dataset.

Model	Channel	Overall Metrics				Per-Class F1-Score				
		Acc	k	MFI	Params/M	W	N1	N2	N3	REM
DeepSleepNet	EEG	82.0 ± 0.2	0.76 ± 0.05	76.9 ± 0.6	21.0	85.0	47.0	86.0	85.0	82.0
MultiChannelSleepNet	EEG-Fpz-Cz, EEG-Pz-Oz, EOG	87.2 ± 0.3	0.82 ± 0.06	81.2 ± 0.5	13.0	92.8	49.1	90.0	89.3	84.8
SleepEEGNet	EEG	84.3 ± 0.2	0.79 ± 0.06	79.7 ± 0.6	2.1	89.2	52.2	89.8	85.1	85.0
TinySleepNet	EEG	85.4 ± 0.3	-	80.5 ± 0.5	1.3	90.1	51.4	88.5	88.3	84.3
SalientSleepNet	EEG-Fpz-Cz, EEG-Pz-Oz, EOG	87.5 ± 0.3	-	83.0 ± 0.6	0.9	92.3	56.2	89.9	87.2	89.2
LMCSleepNet (this work)	EEG-Fpz-Cz, EEG-Pz-Oz, EOG	88.2 ± 0.6	0.84 ± 0.6	82.4 ± 0.4	1.49	93.7	51.5	90.6	90.2	86.6

**Table 5 sensors-25-06065-t005:** Comparison of the proposed LMCSleepNet model with other benchmark models on the SleepEDF-78 dataset.

Model	Channel	Overall Metrics				Per-Class F1-Score				
		Acc	k	MFI	Params/M	W	N1	N2	N3	REM
DeepSleepNet	EEG	78.5	0.73	75.3	21.0	91.0	47.0	81.0	69.0	79.0
MultiChannelSleepNet	EEG-Fpz-Cz, EEG-Pz-Oz, EOG	85.0	0.79	79.6	13.0	94.0	53.0	86.9	81.8	82.6
SleepEEGNet	EEG	82.8	0.73	77.0	2.1	90.3	44.6	85.7	81.6	82.9
TinySleepNet	EEG	83.1	-	78.1	1.3	92.8	51.0	85.3	81.1	80.3
SalientSleepNet	EEG-Fpz-Cz, EEG-Pz-Oz, EOG	84.1	-	79.5	0.9	93.3	54.2	85.8	78.3	85.8
LMCSleepNet (this work)	EEG-Fpz-Cz, EEG-Pz-Oz, EOG	84.1	0.77	77.7	1.49	94.2	48.5	85.8	77.6	82.7

**Table 6 sensors-25-06065-t006:** Lightweight performance validation results of LMCSleepNet model.

ResNet18	CBAM	DSC	MSDC	Overall Metrics				Per-Class F1-Score				
				Acc	k	MFI	Params/M	W	N1	N2	N3	REM
√				87.1	0.82	80.7	11.69	93.2	47.4	89.8	88.6	84.6
√	√			88.0	0.83	81.8	11.71	93.6	49.0	90.5	89.7	86.1
√	√	√		87.3	0.83	80.6	1.48	93.2	44.9	90.1	90.0	85.1
√	√	√	√	88.2	0.84	82.4	1.49	93.7	51.5	90.6	90.2	86.6

## Data Availability

Data available on request from the authors.

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
