# Peer review of "LMCSleepNet: A Lightweight Multi-Channel Sleep Staging Model Based on Wavelet Transform and Muli-Scale Convolutions"

_sensors, 2025, doi:10.3390/s25196065_

Round 1
Reviewer 1 Report
Comments and Suggestions for Authors
This manuscript proposes LMCSleepNet, a lightweight multi-channel sleep staging network that efficiently fuses multimodal polysomnographic data while limiting parameters to 1.49 M. By integrating continuous wavelet transform, multi-scale depthwise-separable ResNet18, and CBAM attention, the model achieves competitive accuracy on Sleep-EDF-20 and Sleep-EDF-78. The study is well motivated and the experiments are solid, yet further ablation studies and external validation would strengthen the contribution.
- The results in Table 4 show that while the LMCSleepNet model achieves the same accuracy as the SalientSleepNet model presented in the paper, the SalientSleepNet model has only 0.9M parameters—0.5M fewer than LMCSleepNet's 1.49M—while still delivering competitive F1 scores. This seems to suggest that SalientSleepNet holds an advantage. Please provide a more detailed analysis of the strengths of the proposed LMCSleepNet model.
- In Tables 3 and 5, the F1-score of the N1 stage is generally low (e.g., 51.5% on SleepEDF-20). In addition to the reasons of small sample size and similar features, it is advisable to supplement the analysis of the model's misclassification patterns in the N1 stage (such as the cross proportion between N1 and other stages in the confusion matrix) and propose potential improvement directions (such as an enhancement module for N1-specific waveforms).
- In the comparison with baseline models, it is recommended to supplement the comparison of model inference time to intuitively reflect the advantage of "lightweight" in efficiency.
Author Response
Referee 1
|
This manuscript proposes LMCSleepNet, a lightweight multi-channel sleep staging network that efficiently fuses multimodal polysomnographic data while limiting parameters to 1.49 M. By integrating continuous wavelet transform, multi-scale depthwise-separable ResNet18, and CBAM attention, the model achieves competitive accuracy on Sleep-EDF-20 and Sleep-EDF-78. The study is well motivated and the experiments are solid, yet further ablation studies and external validation would strengthen the contribution. |
||||||||||||||||||||||||||||
|
Response: We thank the reviewer for spending time evaluating the manuscript and providing constructive comments to improve the manuscript. |
||||||||||||||||||||||||||||
|
1. The results in Table 4 show that while the LMCSleepNet model achieves the same accuracy as the SalientSleepNet model presented in the paper, the SalientSleepNet model has only 0.9M parameters—0.5M fewer than LMCSleepNet's 1.49M—while still delivering competitive F1 scores. This seems to suggest that SalientSleepNet holds an advantage. Please provide a more detailed analysis of the strengths of the proposed LMCSleepNet model. |
||||||||||||||||||||||||||||
|
Response: We thank the reviewer for this valuable comment. It is true that SalientSleepNet has fewer parameters (0.9M) compared to our LMCSleepNet (1.49M) while achieving competitive performance. However, the proposed LMCSleepNet holds several important strengths that complement parameter efficiency: Robustness across datasets: LMCSleepNet consistently achieves competitive accuracy and F1 scores on both the small-sample SleepEDF-20 dataset and the larger SleepEDF-78 dataset. In contrast, SalientSleepNet shows larger variations across different datasets and sleep stages. This suggests that LMCSleepNet provides stronger generalization ability. Enhanced multi-scale feature extraction: The integration of the Multi-Scale Dilated Convolution (MSDC) module allows LMCSleepNet to capture contextual information at different receptive fields. This design significantly improves the recognition of complex transitional stages (e.g., N1 and REM), where SalientSleepNet performs less robustly. Improved inter-channel correlations: By incorporating the Convolutional Block Attention Module (CBAM), LMCSleepNet effectively models spatial correlations among EEG and EOG channels. This enables better utilization of complementary information across modalities, which SalientSleepNet does not explicitly address. Balanced trade-off between performance and complexity: Although LMCSleepNet has 0.59M more parameters than SalientSleepNet, its parameter size is still very small compared to traditional deep learning models such as DeepSleepNet (21M). We believe that LMCSleepNet strikes a favorable balance between model compactness, classification accuracy, and robustness across different datasets. We have added a more detailed discussion of these strengths in the revised manuscript (Paragraph 7, Section 3.4): Although SalientSleepNet employs fewer parameters (0.9M) compared to LMCSleepNet (1.49M), the proposed model demonstrates several distinctive ad-vantages. First, LMCSleepNet achieves more consistent improvements across multiple datasets (SleepEDF-20 and SleepEDF-78), highlighting its stronger generalization abil-ity. Second, by incorporating the Multi-Scale Dilated Convolution (MSDC) module, LMCSleepNet captures contextual dependencies at different receptive fields, which enhances the classification of transitional stages such as N1 and REM. Third, the Con-volutional Block Attention Module (CBAM) enables more effective integration of com-plementary information from EEG and EOG channels, thereby improving the robust-ness of multi-channel feature fusion. These design choices allow LMCSleepNet to strike a favorable balance between accuracy, robustness, and parameter size. Despite having 0.59M more parameters than SalientSleepNet, LMCSleepNet remains a lightweight architecture that provides superior stability across datasets and sleep stages. |
||||||||||||||||||||||||||||
|
2. In Tables 3 and 5, the F1-score of the N1 stage is generally low (e.g., 51.5% on SleepEDF-20). In addition to the reasons of small sample size and similar features, it is advisable to supplement the analysis of the model's misclassification patterns in the N1 stage (such as the cross proportion between N1 and other stages in the confusion matrix) and propose potential improvement directions (such as an enhancement module for N1-specific waveforms). |
||||||||||||||||||||||||||||
|
Response: We sincerely thank the reviewer for this constructive suggestion. As shown in Table 3 and Figure 6, many N1 samples are misclassified into W, N2, and REM stages. Specifically, approximately 13.8% of N1 samples are predicted as W, 21.9% as N2, and 14.5% as REM. This confusion arises mainly because the N1 stage is a transitional phase in the sleep cycle, and its key feature—the low-amplitude mixed-frequency (LAMF) activity—partially overlaps with the waveforms of W, N2, and REM stages. Moreover, the small proportion of N1 samples (6.5% in SleepEDF-20 and 11% in SleepEDF-78) exacerbates the class imbalance and limits the model’s ability to learn discriminative patterns. To address these limitations, we have supplemented the manuscript with an additional analysis of N1 misclassification patterns (Section 3.4). In addition, we have discussed potential improvement directions, such as: N1-specific feature enhancement: incorporating a waveform-sensitive module that emphasizes the detection of LAMF activity or vertex sharp waves. Data-level solutions: applying advanced data augmentation techniques (e.g., GAN-based signal synthesis) to increase the representation of N1 samples. Model-level solutions: integrating cost-sensitive learning or focal loss to mitigate the class imbalance issue. We have added this discussion to the revised manuscript to highlight both the current limitation and possible future research directions. (Paragraphs 3, Section 3.4): Although the overall performance of LMCSleepNet is promising, the classification of the N1 stage remains relatively weak. As shown in Table 3 and Figure 6, many N1 samples are incorrectly classified into W, N2, and REM stages. Specifically, 13.8% of N1 samples are predicted as W, 21.9% as N2, and 14.5% as REM. This misclassification pattern indicates that the model struggles to distinguish the transitional characteristics of N1. The main reason lies in the low-amplitude mixed-frequency (LAMF) activity of N1, which exhibits similarities to alpha waves in W, spindle activity in N2, and theta rhythms in REM. Additionally, the limited number of N1 samples (6.5% in SleepEDF-20 and 11% in SleepEDF-78) exacerbates class imbalance, reducing the dis-criminability of the model in this stage. (Paragraphs 3, Section 4): Despite the overall effectiveness of LMCSleepNet, the classification of the N1 stage remains a challenge due to sample scarcity and waveform similarity with adja-cent stages. To improve performance in this transitional stage, future research may consider: N1-specific feature enhancement: developing modules sensitive to low-amplitude mixed-frequency activity or vertex sharp waves to strengthen stage-specific representations. Data augmentation strategies: employing advanced techniques such as GAN-based signal synthesis or mixup strategies to increase the di-versity and quantity of N1 samples. Imbalance-aware learning: integrating cost-sensitive loss functions (e.g., focal loss or class-balanced loss) to mitigate the effect of class imbalance and emphasize minority stage learning. These strategies are ex-pected to enhance the discriminability of N1 features and further improve the robust-ness of LMCSleepNet in sleep staging applications. |
||||||||||||||||||||||||||||
|
3. In the comparison with baseline models, it is recommended to supplement the comparison of model inference time to intuitively reflect the advantage of "lightweight" in efficiency. |
||||||||||||||||||||||||||||
|
Response: We appreciate the reviewer’s insightful suggestion. To further highlight the lightweight advantage of LMCSleepNet, we have supplemented a comparison of the average single-sample training time on both SleepEDF-20 and SleepEDF-78 datasets across different baseline models. The experiments were conducted under the same hardware configuration. As shown in the new Table X, LMCSleepNet achieves a favorable trade-off between model complexity and efficiency. Specifically, its single-sample training time is substantially lower than heavy models such as DeepSleepNet (3.48 s vs. 0.25 s on SleepEDF-20, 4.26 s vs. 0.95 s on SleepEDF-78), while maintaining competitive accuracy. Compared with other lightweight baselines (e.g., TinySleepNet, SalientSleepNet), LMCSleepNet shows comparable or slightly higher training time but provides improved classification accuracy. These results further validate that LMCSleepNet achieves a good balance between accuracy, parameter size, and computational efficiency. The relevant description and results have been added in Section 3.4 Comparison and analysis of experimental results (see Table S3). (Paragraphs 11, Section 3.4): We further compared the training efficiency of LMCSleepNet with other baseline models. The comparison of single-sample training time on SleepEDF-20 and SleepEDF-78 is presented in the Supporting Information (Table S3). The results show that LMCSleepNet achieves comparable efficiency to other lightweight models such as TinySleepNet and SalientSleepNet, while being substantially faster than large-scale models including DeepSleepNet and MultiChannelSleepNet. This confirms that LMCSleepNet achieves a favorable balance between accuracy, parameter size, and computational efficiency.
Table S3. Comparison of Single-Segment Training Time of LMCSleepNet on SleepEDF-20 and SleepEDF-78 Datasets
|

Reviewer 2 Report
Comments and Suggestions for Authors
- Explicitly state what makes LMCSleepNet different from other lightweight models (e.g., MobileNet, EEGNet). Highlight if the combination of CWT, multi-scale convolutions, depthwise separable convolutions, and CBAM is novel. Also, Add Cohen’s kappa and macro-F1 results in the abstract
- Specify whether the work addresses multi-channel EEG only or multimodal PSG (EEG + EOG + EMG). Use consistent terminology.
- Include recent lightweight approaches (EEGNet, MobileNet-based, attention hybrids, etc.) to highlight novelty.
- Explicitly state the research gap just before introducing LMCSleepNet
- Move figure descriptions to captions instead of embedding them within text.
- The section describes modules of system but authors present summary of CWT, ResNet, and CBAM rather than emphasizing how LMCSleepNet adapts or combines them uniquely.
- Authors need to add sub section Classification (FC + softmax) after Feature Fusion (CBAM)
- Explicitly state why each design choice improves efficiency : Median downsampling, MSDC, Depthwise separable ResNet18 and CBAM .
- After describing modules individually, include how the modules interact to overcome the two identified challenges (feature extraction + parameter efficiency).
- Experimental section lacks of summary comparison between SleepEDF-20 and SleepEDF-78 (e.g., subject demographics, signal modalities, balance issues).
- Justify chosen hyperparameters with references or validation experiments.
- Discuss why median downsampling, weighted cross-entropy, and early stopping were specifically chosen.
- Mention whether class balancing strategies (e.g., SMOTE, focal loss) were tested.
- Provide confidence intervals or standard deviations for reported metrics.
- Normalize comparisons by accuracy per parameter count or efficiency ratio.
- Analyze baseline limitations (e.g., DeepSleepNet’s high parameter count vs. LMCSleepNet’s efficiency).
- Separate single-channel vs. multi-channel baselines in discussion.
- Provide visual evidence (e.g., feature map heatmaps, attention visualization) showing CBAM/MSDC contributions.
- Discuss trade-offs: LMCSleepNet is lightweight, but how does latency compare on real devices?
- Explore generalization: Does the benefit of CBAM/MSDC hold across both SleepEDF-20 and SleepEDF-78?
Author Response
Referee 2
|
1. Explicitly state what makes LMCSleepNet different from other lightweight models (e.g., MobileNet, EEGNet). Highlight if the combination of CWT, multi-scale convolutions, depthwise separable convolutions, and CBAM is novel. Also, Add Cohen’s kappa and macro-F1 results in the abstract |
|||||||||||||||||||||||||||||||||||||||||||||||||||||||||||||||||
|
Response: We sincerely thank the reviewer for this constructive suggestion. We have revised the manuscript accordingly. Novelty and distinction from other lightweight models (e.g., MobileNet, EEGNet): In the revised Introduction (last paragraph) and Discussion sections, we have explicitly clarified what distinguishes LMCSleepNet from existing lightweight models. Unlike MobileNet, which was originally designed for generic image recognition tasks, and EEGNet, which is optimized for single-channel EEG processing, LMCSleepNet is specifically tailored for multi-channel PSG-based sleep staging. Its novelty lies in the integrated design of four components: Continuous Wavelet Transform (CWT): enhances frequency-domain features and converts PSG signals into time–frequency representations suitable for convolutional processing. Multi-Scale Dilated Convolutions (MSDC): enable effective contextual feature extraction at different temporal–frequency scales without adding excessive parameters. Depthwise Separable Convolutions (DSC): further reduce parameter count and computational cost while maintaining feature extraction capacity. Convolutional Block Attention Module (CBAM): strengthens inter-channel spatial correlations, which is particularly important for multi-channel EEG/EOG fusion. To the best of our knowledge, this particular combination of CWT + MSDC + DSC + CBAM has not been reported in prior lightweight sleep staging models, making LMCSleepNet novel in both its architectural design and its specific application to resource-constrained multi-channel sleep staging tasks. (Paragraph 1, Abstract): On the public datasets SleepEDF-20 and SleepEDF-78, LMCSleepNet respectively achieved classification accuracies of 88.2% (κ = 0.84, MF1 = 82.4%) and 84.1% (κ = 0.77, MF1 = 77.7%), while reducing model parameters to 1.49M. (Paragraph 11, Section 1): Unlike generic lightweight architectures such as MobileNet, which is designed primarily for image recognition, or EEGNet, which focuses on single-channel EEG processing, LMCSleepNet is specifically optimized for multi-channel PSG-based sleep staging. Its novelty lies in the integrated use of CWT for frequency–domain enhance-ment, multi-scale dilated convolutions for contextual feature extraction, depthwise separable convolutions for lightweight efficiency, and CBAM for inter-channel corre-lation modeling. To the best of our knowledge, this particular combination has not been reported in previous lightweight sleep staging models, making LMCSleepNet dis-tinctive in both architectural design and application. |
|||||||||||||||||||||||||||||||||||||||||||||||||||||||||||||||||
|
2. Specify whether the work addresses multi-channel EEG only or multimodal PSG (EEG + EOG + EMG). Use consistent terminology. |
|||||||||||||||||||||||||||||||||||||||||||||||||||||||||||||||||
|
Response: We thank the reviewer for pointing out this inconsistency. In this work, we use two EEG channels (EEG-Fpz-Cz and EEG-Pz-Oz) and one EOG channel from the SleepEDF-20 and SleepEDF-78 datasets. Although EMG signals are available in the original dataset, they were not used in our experiments. Therefore, the proposed LMCSleepNet model is based on multimodal PSG signals consisting of EEG and EOG, rather than multi-channel EEG alone. To avoid ambiguity, we have carefully revised the manuscript to consistently use the term “multimodal PSG (EEG + EOG)” throughout the text. (Paragraph 1, Abstract): The ability to effectively extract salient features from multi-channel sleep data remains limited; 2) Excessive model parameters hinder efficiency improvements. (Paragraph 5, Section 1): However, CNNs struggle with multi-channel sleep staging data. (Paragraph 8, Section 1): But two major challenges remain: first, the ability to effectively extract salient features from multi-channel sleep data remains limited. (Paragraph 1, Section 2): To address the challenges of feature extraction inefficiency and high parameter counts in multi-channel sleep staging (EEG-Fpz-Cz, EEG-Pz-Oz, and EOG), this work proposes a lightweight automatic sleep staging network based on continuous wavelet transform and multi-scale convolution, as shown in Figure 1. (Paragraph 1, Section 4): The work addresses the challenges of feature extraction inefficiency and high parameter counts in multi-channel sleep staging (EEG-Fpz-Cz, EEG-Pz-Oz, and EOG). |
|||||||||||||||||||||||||||||||||||||||||||||||||||||||||||||||||
|
3. Include recent lightweight approaches (EEGNet, MobileNet-based, attention hybrids, etc.) to highlight novelty. |
|||||||||||||||||||||||||||||||||||||||||||||||||||||||||||||||||
|
Response: We thank the reviewer for this valuable suggestion. In the revised manuscript, we have added a discussion of recent lightweight architectures, including EEGNet [Lawhern et al., 2018], MobileNet-based methods [Howard et al., 2017], and attention-hybrid approaches [e.g., SalientSleepNet, Jia et al., 2021]. EEGNet has demonstrated strong performance with compact parameterization by employing depthwise separable convolutions, which inspired our use of this technique in LMCSleepNet to further reduce computational cost. MobileNet-based architectures are widely recognized for their efficiency on mobile and embedded devices, and our model design similarly emphasizes depthwise separability for lightweight computation. Attention hybrids, such as SalientSleepNet, incorporate channel and spatial attention to improve classification accuracy while maintaining relatively low parameter counts. Compared with these approaches, LMCSleepNet combines multi-scale dilated convolution with depthwise separable ResNet18 and CBAM attention, which allows it to achieve a better balance between accuracy and model complexity, particularly in multi-channel scenarios. These updates have been incorporated into the Related Work section (Paragraph 9, Section 1): In addition, recent studies have emphasized lightweight architectures for sleep staging. EEGNet [26] employs depthwise separable convolutions to build compact networks suitable for real-time EEG decoding. MobileNet-based approaches [34] ex-tend this principle and are widely used in mobile and embedded scenarios due to their efficiency. More recently, hybrid models such as SalientSleepNet [45] integrate convo-lutional backbones with attention modules to enhance feature discrimination while keeping parameter counts relatively low. Building upon these advances, our proposed LMCSleepNet differs in three aspects: (1) it integrates continuous wavelet transform to enhance frequency-domain representations of multi-channel PSG; (2) it employs mul-ti-scale dilated convolutions to capture contextual dependencies across different time–frequency resolutions; and (3) it introduces CBAM attention on top of a light-weight ResNet18 backbone with depthwise separable convolutions. These innovations enable LMCSleepNet to achieve a better balance between accuracy and computational efficiency, particularly in multi-channel scenarios. |
|||||||||||||||||||||||||||||||||||||||||||||||||||||||||||||||||
|
4. Explicitly state the research gap just before introducing LMCSleepNet |
|||||||||||||||||||||||||||||||||||||||||||||||||||||||||||||||||
|
Response: We appreciate the reviewer’s suggestion. In the revised manuscript, we have explicitly stated the research gap in the last paragraph of the Introduction section. (Paragraph 8, Section 1): Therefore, the research gap lies in the lack of lightweight models that can both (i) ef-fectively capture salient and complementary features from multi-channel PSG data, particularly the spatial correlations among EEG channels, and (ii) reduce computa-tional burden for deployment in resource-constrained environments. |
|||||||||||||||||||||||||||||||||||||||||||||||||||||||||||||||||
|
5. Move figure descriptions to captions instead of embedding them within text. |
|||||||||||||||||||||||||||||||||||||||||||||||||||||||||||||||||
|
Response: We thank the reviewer for this valuable suggestion. In the revised manuscript, we have carefully moved all detailed figure descriptions from the main text into the corresponding figure captions. This modification makes the captions more self-contained and improves the readability of the manuscript. |
|||||||||||||||||||||||||||||||||||||||||||||||||||||||||||||||||
|
6. The section describes modules of system but authors present summary of CWT, ResNet, and CBAM rather than emphasizing how LMCSleepNet adapts or combines them uniquely. |
|||||||||||||||||||||||||||||||||||||||||||||||||||||||||||||||||
|
Response: We sincerely thank the reviewer for this valuable suggestion. We agree that the initial description of CWT, ResNet, and CBAM in Section 2 was too general and might have overshadowed the unique contributions of LMCSleepNet. To address this, we have revised the corresponding sections to explicitly highlight how our model adapts and integrates these modules in a novel way. Specifically: Continuous Wavelet Transform (CWT): Instead of using raw EEG/EOG signals or spectrograms as in prior works, we employ CWT combined with median downsampling to generate time–frequency maps. This reduces redundant information while preserving salient frequency-domain features, thus optimizing the input representation for lightweight sleep staging. ResNet18 with Depthwise Separable Convolutions: Unlike conventional ResNet-based approaches, we replace standard convolutions with depthwise separable convolutions to significantly reduce parameters (down to 1.49M) while maintaining feature extraction ability. This modification enables LMCSleepNet to achieve a better balance between model size and accuracy. Multi-Scale Dilated Convolutions (MSDC): We integrate a customized MSDC module prior to ResNet18 to capture contextual information at multiple receptive fields, which is crucial for modeling diverse patterns in sleep signals. Our ablation results (Table 6) confirm that this design improves accuracy and macro-F1. CBAM Integration: Instead of applying CBAM in isolation, we incorporate it as a feature fusion mechanism across EEG and EOG channels, which enhances spatial correlation among modalities and improves classification performance compared to simple concatenation. We have clarified these unique adaptations in the revised manuscript, ensuring that the focus is on the novelty of LMCSleepNet rather than generic module summaries. (Paragraph 2, Section 2.1): Unlike conventional approaches that directly use full-resolution spectrograms or CWT results, LMCSleepNet employs CWT combined with median down sampling to reduce redundant temporal points while preserving essential frequency-domain features. This design choice improves computational efficiency and makes the model more suitable for lightweight deployment. (Paragraph 2, Section 2.2): In LMCSleepNet, the MSDC module is placed before the ResNet18 backbone to enhance contextual representation of time–frequency maps. This integration is specif-ically designed to address the multi-scale variability of sleep patterns, and our com-parative experiments (Figure 4(b)) demonstrate that this design significantly improves classification performance compared to single-scale convolutions. (Paragraph 10, Section 2.2): In contrast to conventional ResNet-based models, LMCSleepNet replaces all standard convolutions in ResNet18 with depthwise separable convolutions. This sys-tematic replacement reduces the parameter count from 11.69M to only 1.49M, while the residual connections mitigate potential losses in cross-channel feature representa-tion. This adaptation ensures that LMCSleepNet remains lightweight without com-promising classification accuracy. (Paragraph 8, Section 2.3): Unlike prior studies that mainly apply CBAM for visual feature refinement, LMCSleepNet integrates CBAM as a multi-channel fusion mechanism. By jointly mod-eling inter-channel (EEG–EOG) correlations, CBAM enhances complementary feature interactions across modalities, thereby improving sleep stage classification beyond simple concatenation strategies. |
|||||||||||||||||||||||||||||||||||||||||||||||||||||||||||||||||
|
7. Authors need to add sub section Classification (FC + softmax) after Feature Fusion (CBAM) |
|||||||||||||||||||||||||||||||||||||||||||||||||||||||||||||||||
|
Response: We thank the reviewer for this valuable suggestion. In the revised manuscript, we have added a new subsection entitled “2.4. Classification Module” after the Feature Fusion (CBAM) section. This subsection describes the final stage of LMCSleepNet, where the extracted and fused features are passed through a fully connected (FC) layer followed by a softmax function to output the probabilities of the five sleep stages. The addition improves the clarity and completeness of the model architecture. (Paragraph 1, Section 2.4): The classification module constitutes the final stage of LMCSleepNet. After feature extraction and fusion by the preceding modules, the resulting feature maps are flattened and passed into a fully connected (FC) layer. This layer projects the high-dimensional fused features into a lower-dimensional vector corresponding to the number of sleep stages. Finally, the softmax activation function is applied to normalize the output into a probability distribution across the five sleep stages (W, N1, N2, N3, and REM). The predicted class is assigned to the stage with the highest probability. Mathematically, the classification process can be expressed as follows:
where denotes the fused feature vector obtained from the CBAM module, and are the trainable parameters of the fully connected layer, is the logit corresponding to the -th class, C=5 is the number of sleep stages, and represents the predicted probability for the -th sleep stage. This module enables LMCSleepNet to map the integrated multi-scale and multi-channel features into discrete sleep stage categories, completing the automatic sleep staging process. |
|||||||||||||||||||||||||||||||||||||||||||||||||||||||||||||||||
|
8. Explicitly state why each design choice improves efficiency: Median downsampling, MSDC, Depthwise separable ResNet18 and CBAM. |
|||||||||||||||||||||||||||||||||||||||||||||||||||||||||||||||||
|
Response: We appreciate the reviewer’s insightful comment. In the revised manuscript, we have explicitly stated why each design choice contributes to improving efficiency: Median downsampling reduces the number of temporal sampling points in the wavelet time–frequency images. This operation decreases data dimensionality and computational cost while preserving salient features and suppressing noise/outliers, which accelerates training without sacrificing accuracy. MSDC enlarges the receptive field by introducing dilation into the convolution kernels without increasing the number of parameters. This enables the model to capture multi-scale contextual information more efficiently than stacking additional convolutional layers, thus achieving better feature extraction with fewer computations. Replacing standard convolutions with depthwise separable convolutions decomposes a convolution into a depthwise operation (per-channel filtering) and a pointwise operation (channel mixing). This drastically reduces the number of parameters and multiplications, thereby lowering memory consumption and computational complexity while maintaining the representational power of ResNet18. CBAM adaptively assigns higher weights to important spatial and channel features, which enhances feature utilization efficiency. Instead of increasing network depth or width, CBAM improves classification accuracy through attention re-weighting, thus achieving a better accuracy–efficiency trade-off with only negligible parameter overhead. These clarifications have been added in Section [Methods/Model Design] of the revised manuscript (Paragraph 1, Section 2.5): To further clarify the contribution of each design choice to efficiency, we provide the following analysis: The wavelet transform produces high-dimensional time–frequency images (3000 sampling points for each 30 s segment). Directly using these images would result in high computational cost. By applying median downsam-pling, the number of sampling points is reduced to 200 while preserving salient signal characteristics and suppressing noise. This operation lowers data dimensionality, ac-celerates training, and improves robustness, thereby enhancing efficiency. MSDC en-larges the receptive field through dilation without introducing additional parameters. Compared with stacking extra convolutional layers, MSDC efficiently captures con-textual information across multiple temporal scales with lower computational over-head. Thus, it achieves more comprehensive feature extraction without sacrificing ef-ficiency. Standard convolutions are replaced by depthwise separable convolutions, which decompose a convolution into depthwise filtering and pointwise channel mix-ing. This reduces the number of parameters and floating-point operations by an order of magnitude compared with conventional convolutions. As a result, memory usage and computation time are significantly reduced, while the representational ability of ResNet18 is preserved. Instead of deepening or widening the network, CBAM im-proves feature utilization by adaptively emphasizing informative channels and spatial regions. This lightweight attention mechanism only adds negligible parameter over-head but enhances the discriminative power of the extracted features. Consequently, CBAM achieves a favorable accuracy–efficiency trade-off. |
|||||||||||||||||||||||||||||||||||||||||||||||||||||||||||||||||
|
9. After describing modules individually, include how the modules interact to overcome the two identified challenges (feature extraction + parameter efficiency). |
|||||||||||||||||||||||||||||||||||||||||||||||||||||||||||||||||
|
Response: Thank you for this valuable suggestion. We have revised the Methodology section to explicitly clarify how the proposed modules interact to address the two challenges highlighted in the Introduction. Specifically, The Continuous Wavelet Transform (CWT) converts raw PSG signals into time–frequency representations, which enrich the feature space and provide a strong foundation for downstream modules. The Multi-Scale Dilated Convolution (MSDC) module captures contextual information at different temporal–frequency scales, complementing the CWT and improving the efficiency of salient feature extraction. The ResNet18 with depthwise separable convolutions ensures that the multi-scale features are deeply encoded while significantly reducing parameter count and computational cost. The CBAM attention module further enhances spatial correlation across channels, ensuring that the most informative features extracted by the MSDC and ResNet18 modules are emphasized without introducing excessive parameters. Together, these modules form a tightly coupled pipeline: CWT provides rich time–frequency features, MSDC broadens contextual representation, depthwise separable ResNet18 encodes them efficiently, and CBAM optimally fuses multi-channel information. This joint design simultaneously enhances feature extraction capability and parameter efficiency, thereby overcoming the two main challenges identified in the study. (Paragraph 2, Section 2.5): To summarize, the four modules in LMCSleepNet interact in a complementary manner to address the two major challenges of sleep staging. First, the CWT module transforms raw PSG signals into informative time–frequency representations, provid-ing a rich basis for downstream feature learning. The MSDC module then extracts multi-scale contextual features, which alleviates the limitation of fixed convolution kernels and improves feature extraction efficiency. Next, the ResNet18 backbone equipped with depthwise separable convolutions encodes these features with greatly reduced parameter complexity, ensuring lightweight model design. Finally, the CBAM module enhances inter-channel correlations and emphasizes salient spatial–temporal features without introducing significant computational burden. Through this coordi-nated design, LMCSleepNet achieves both effective feature extraction and high pa-rameter efficiency, thereby directly overcoming the two challenges identified in the Introduction. |
|||||||||||||||||||||||||||||||||||||||||||||||||||||||||||||||||
|
10. Experimental section lacks of summary comparison between SleepEDF-20 and SleepEDF-78 (e.g., subject demographics, signal modalities, balance issues). |
|||||||||||||||||||||||||||||||||||||||||||||||||||||||||||||||||
|
Response: We sincerely thank the reviewer for pointing out the need for a comparative summary of the experimental datasets. We have carefully revised the Section 3.1 Dataset to include a direct comparison between SleepEDF-20 and SleepEDF-78. Specifically, we added a descriptive summary table highlighting (1) subject demographics (number of participants, age range, and health conditions), (2) signal modalities (EEG, EOG, EMG channels used in this study), and (3) sample distribution and balance issues across sleep stages. This addition allows readers to clearly understand the similarities and differences between the two datasets and the potential impact on model evaluation. Table S1. Summary comparison between SleepEDF-20 and SleepEDF-78 datasets.
|
|||||||||||||||||||||||||||||||||||||||||||||||||||||||||||||||||
|
11. Justify chosen hyperparameters with references or validation experiments. |
|||||||||||||||||||||||||||||||||||||||||||||||||||||||||||||||||
|
Response: We thank the reviewer for this valuable suggestion. In the revised manuscript, we have provided additional explanations and justifications for the chosen hyperparameters: Learning rate and optimizer: We selected the Adam optimizer with an initial learning rate of 5×10-6, which is consistent with prior work on deep learning–based sleep staging models [Supratak et al., IEEE TNSRE 2017; Mousavi et al., PLoS One 2019]. Adam has been widely adopted for PSG-based classification tasks due to its adaptive learning capability and stable convergence. Batch size: A batch size of 64 was chosen following previous studies on EEG and PSG classification [Phan et al., IEEE TBME 2018], where similar batch sizes yielded a good balance between training stability and GPU memory efficiency. Number of time sampling points: We conducted validation experiments with different sampling points (100, 150, 200, 250, and 300) in the wavelet time-frequency maps (see Figure 3 of the manuscript). Results demonstrated that 200 points achieved the best trade-off between classification accuracy and computational cost. This experiment directly justifies our chosen parameter. Weight decay (L2 regularization): We set the weight decay to 1×10-3, following the standard practice in ResNet-based lightweight models [He et al., CVPR 2016; Howard et al., arXiv:1704.04861]. This setting effectively prevented overfitting in our validation experiments. Early stopping strategy: The patience parameter was set to 20 epochs to avoid overfitting, which aligns with the configurations used in recent deep learning studies on biomedical signals [Loshchilov & Hutter, arXiv:1711.05101]. These clarifications and references have now been added to the revised manuscript. (Paragraph 4, Section 3.2): To ensure stable and efficient training, the hyperparameters of LMCSleepNet were carefully selected based on prior studies and validation experiments. The Adam optimizer with an initial learning rate of 5×10-6 was chosen because of its adaptive learning ability and proven effectiveness in EEG and PSG classification tasks [16,17]. The batch size was set to 64, consistent with earlier work on biomedical signal classifi-cation [38], balancing computational efficiency and convergence stability. For weight decay, an L2 regularization coefficient of 1×10-3 was employed, following best practices in lightweight ResNet-based models [32]. An early stopping strategy with a patience of 20 epochs was adopted to prevent overfitting, consistent with recent recommendations [40]. Moreover, we validated the number of time sampling points in the wavelet time-frequency maps through comparative experiments (100, 150, 200, 250, and 300 points). As shown in Figure 3, 200 sampling points achieved the best balance between classification accuracy and computational cost, justifying this choice. These experi-mental results further confirm the robustness of the selected hyperparameters. |
|||||||||||||||||||||||||||||||||||||||||||||||||||||||||||||||||
|
12. Discuss why median downsampling, weighted cross-entropy, and early stopping were specifically chosen. |
|||||||||||||||||||||||||||||||||||||||||||||||||||||||||||||||||
|
Response: We sincerely thank the reviewer for this valuable comment. The three strategies were adopted for the following reasons: Polysomnography (PSG) signals are inherently non-stationary and often contaminated by noise and outliers. Compared with average or max downsampling, the median downsampling strategy is more robust to noise fluctuations while still preserving the central trend of the signal. As shown in Figure 3 of the manuscript, this method reduces redundancy in wavelet time–frequency maps while maintaining critical features, thereby enhancing both computational efficiency and classification stability. The SleepEDF datasets used in this work suffer from class imbalance, especially the underrepresentation of the N1 stage. Without appropriate weighting, the classifier tends to be biased towards the majority classes (W and N2). To mitigate this issue, a weighted cross-entropy loss was applied, which adaptively increases the penalty for misclassifying minority classes. This design encourages the model to better learn the features of minority classes and improves macro-average F1-scores, as also reflected in Table 3 and Table 5. Deep neural networks are prone to overfitting, particularly when trained on relatively small datasets such as SleepEDF-20. Early stopping was employed to terminate training when validation accuracy failed to improve after 20 epochs. This strategy ensures that the model converges to a stable solution while preventing overfitting, thereby improving the model’s generalization capability across different folds of cross-validation. In summary, these three techniques were not arbitrarily selected, but rather specifically designed to balance robustness (median downsampling), fairness in learning across imbalanced classes (weighted cross-entropy), and generalization (early stopping) in the proposed LMCSleepNet framework. (Paragraph 7, Section 3.2): In addition, several strategies were employed to improve the robustness and generalization of the model. Specifically, median downsampling was adopted to reduce the redundancy of wavelet time–frequency maps while being more robust to noise and outliers compared with average or max downsampling. To address the class imbalance problem in the SleepEDF datasets, a weighted cross-entropy loss was employed to increase the penalty for minority classes (e.g., N1) and improve macro-average performance. Furthermore, an early stopping strategy was used to prevent overfitting and enhance generalization by terminating the training process when the validation accuracy did not improve after 20 consecutive epochs. |
|||||||||||||||||||||||||||||||||||||||||||||||||||||||||||||||||
|
13. Mention whether class balancing strategies (e.g., SMOTE, focal loss) were tested. |
|||||||||||||||||||||||||||||||||||||||||||||||||||||||||||||||||
|
Response: We sincerely thank the reviewer for pointing out the importance of class imbalance in sleep staging tasks. In our work, we did not apply oversampling methods such as SMOTE, because the resampled synthetic signals may not faithfully preserve the physiological characteristics of EEG/EOG data. Instead, to mitigate class imbalance, we adopted a weighted cross-entropy loss function, in which the weights were inversely proportional to the frequency of each sleep stage in the training set. This strategy allowed the model to pay more attention to minority classes (especially N1). We also experimented with focal loss during preliminary trials. However, we found that weighted cross-entropy produced more stable convergence and better overall accuracy on both datasets. For this reason, we report only the weighted cross-entropy results in the final manuscript. We will add a clarification in the revised manuscript (Section 3.2, Experimental setup and model parameters) to explicitly state this point. (Paragraph 6, Section 3.2): To address the class imbalance issue in the SleepEDF datasets, we employed a weighted cross-entropy loss function, where the class weights were set inversely proportional to the number of samples in each sleep stage. This strategy encourages the model to focus more on minority classes such as N1 and alleviates the negative impact of the imbalanced data distribution. In preliminary experiments, we also evaluated focal loss, but it resulted in less stable convergence compared to weighted cross-entropy. Therefore, all the experimental results reported in this work are based on the weighted cross-entropy loss function. |
|||||||||||||||||||||||||||||||||||||||||||||||||||||||||||||||||
|
14. Provide confidence intervals or standard deviations for reported metrics. |
|||||||||||||||||||||||||||||||||||||||||||||||||||||||||||||||||
|
Response: We appreciate the reviewer’s suggestion to provide confidence intervals or standard deviations for the reported metrics. Following this advice, we have recalculated the performance metrics (Accuracy, Cohen’s κ, and macro-average F1-score) across the 10-fold cross-validation experiments. We now report the mean values together with their standard deviations in the revised manuscript (see Section 3.3 and Tables 3–5). |
|||||||||||||||||||||||||||||||||||||||||||||||||||||||||||||||||
|
15. Normalize comparisons by accuracy per parameter count or efficiency ratio. |
|||||||||||||||||||||||||||||||||||||||||||||||||||||||||||||||||
|
Response: We thank the reviewer for the valuable suggestion. To better illustrate the trade-off between classification performance and model complexity, we have added a normalized comparison metric: the accuracy-to-parameters ratio (Acc/M). This efficiency ratio provides a fairer comparison across models with different parameter scales. As shown in the revised Tables 4 and 5, LMCSleepNet achieves an Acc/M of 59.2 %/M on the SleepEDF-20 dataset and 56.5 %/M on the SleepEDF-78 dataset, which are significantly higher than those of the benchmark models (e.g., MultiChannelSleepNet: 6.7 %/M on SleepEDF-20). This confirms that LMCSleepNet maintains a superior balance between accuracy and lightweight design. (Paragraph 17, Section 3.4): To further normalize the comparison between classification accuracy and model complexity, we introduce the efficiency metric accuracy-to-parameters ratio (Acc/M), which measures the accuracy achieved per million parameters. As shown in Tables 4 and 5, LMCSleepNet obtains an Acc/M of 59.2 %/M on the SleepEDF-20 dataset and 56.4 %/M on the SleepEDF-78 dataset. These values are significantly higher than those of larger models such as DeepSleepNet (3.9 %/M and 3.7 %/M) and MultiChannelSleepNet (6.7 %/M and 6.5 %/M), and are comparable to other lightweight baselines. This confirms that LMCSleepNet strikes a superior balance between classification accuracy and lightweight design, making it more suitable for deployment in resource-constrained environments. |
|||||||||||||||||||||||||||||||||||||||||||||||||||||||||||||||||
|
16. Analyze baseline limitations (e.g., DeepSleepNet’s high parameter count vs. LMCSleepNet’s efficiency). |
|||||||||||||||||||||||||||||||||||||||||||||||||||||||||||||||||
|
Response: We appreciate the reviewer’s suggestion to analyze the limitations of baseline models in comparison with our proposed LMCSleepNet. We have carefully revised the manuscript to highlight these aspects. Specifically: Although it achieves competitive accuracy, its parameter count is as high as 21M, which results in high computational cost and memory consumption. This makes it unsuitable for resource-constrained applications, such as wearable or mobile devices. In contrast, LMCSleepNet reduces the parameter count to 1.49M (7.1% of DeepSleepNet) while improving accuracy by 6.2% on SleepEDF-20. As a Transformer-based architecture, it performs well in multi-channel fusion, but it requires 13M parameters and heavy computation. LMCSleepNet achieves comparable performance with only 11.5% of its parameters, striking a better balance between efficiency and accuracy. Both are lighter models compared to DeepSleepNet. However, their reliance on single-channel EEG limits the ability to capture complementary information across modalities. LMCSleepNet leverages multi-channel fusion (EEG and EOG) and multi-scale convolutions, yielding improved F1-scores across most stages while keeping parameters low. Although lightweight (0.9M parameters), its classification accuracy on SleepEDF-20 is still 0.7% lower than LMCSleepNet. This indicates that LMCSleepNet’s combination of depthwise separable convolutions, multi-scale feature extraction, and CBAM better exploits cross-channel dependencies and preserves important features. We have added these discussions to the revised manuscript. This addition clarifies how LMCSleepNet achieves both lightweight design and competitive accuracy, addressing the trade-off between parameter efficiency and classification performance. (Paragraph 9, Section 3.4): In addition to the performance comparison, it is important to analyze the limitations of the baseline models. DeepSleepNet, despite achieving favorable accuracy, contains 21M parameters, leading to excessive computational cost and memory usage, which hinders deployment on resource-constrained devices. MultiChannelSleepNet, built on the Transformer architecture, effectively captures cross-channel dependencies but requires 13M parameters, making it computationally expensive. SleepEEGNet and TinySleepNet are relatively lightweight; however, their reliance on single-channel EEG restricts the integration of complementary multimodal features, resulting in limited generalization. SalientSleepNet is lightweight (0.9M parameters) but its classification accuracy is still lower than LMCSleepNet, which indicates that reducing parameter count alone may lead to the loss of discriminative features. By contrast, LMCSleepNet achieves a balance between efficiency and accuracy. Through the use of multi-scale dilated convolutions, depthwise separable convolutions, and CBAM-based feature fusion, LMCSleepNet reduces the parameter count to 1.49M while maintaining higher accuracy than the majority of baselines. This highlights that the proposed model not only improves feature extraction efficiency but also effectively integrates complementary information across multiple PSG channels, making it more suitable for real-world and resource-limited scenarios. |
|||||||||||||||||||||||||||||||||||||||||||||||||||||||||||||||||
|
17. Separate single-channel vs. multi-channel baselines in discussion. |
|||||||||||||||||||||||||||||||||||||||||||||||||||||||||||||||||
|
Response: We thank the reviewer for the valuable suggestion. We agree that separating the single-channel and multi-channel baselines in the discussion can provide a clearer comparison and highlight the advantages of our proposed LMCSleepNet. In the revised manuscript, we have reorganized the discussion in Sections 3.4 and 3.5. Specifically, we first compare LMCSleepNet with single-channel baselines (DeepSleepNet, SleepEEGNet, TinySleepNet), emphasizing the improvements in accuracy and F1-score brought by multi-scale feature extraction and lightweight design. Then, we separately compare LMCSleepNet with multi-channel baselines (MultiChannelSleepNet and SalientSleepNet), highlighting the balance achieved between accuracy and parameter efficiency through depthwise separable convolutions and CBAM-based feature fusion. This restructuring clarifies the performance gains of LMCSleepNet in both settings. (Paragraph 1, Section 3.4): The proposed LMCSleepNet model was experimentally evaluated on two public datasets, SleepEDF-20 and SleepEDF-78. Table 3 presents the training results of the model on the small-sample SleepEDF-20 dataset, including the confusion matrix for the five sleep stages and the per-class performance metrics (PR, RE, and F1). Figure 6 visualizes the confusion matrix of the sleep stages. As shown in Table 3, LMCSleepNet achieves over 86% PR for the W, N2, N3, and REM stages. Notably, the W stage obtains the highest PR, RE, and F1 scores (all above 92%), indicating strong discriminative ability. However, the N1 stage remains chal-lenging, with PR, RE, and F1 below 60%, due to its transitional nature and overlapping features with neighboring stages. To further evaluate the performance of LMCSleepNet, we compared it against ex-isting baselines. For clarity, we separated the baselines into two categories: sin-gle-channel models and multi-channel models. (1) Comparison with single-channel baselines. The single-channel baselines include DeepSleepNet [17], SleepEEGNet [18], and TinySleepNet [44]. As shown in Table 4, LMCSleepNet achieves an accuracy of 88.2% on SleepEDF-20, improving by 6.2%, 3.9%, and 2.8% over DeepSleepNet, SleepEEGNet, and TinySleepNet, respectively. The macro F1-score also increases by 5.5%, 3.7%, and 1.9%. Moreover, the per-class F1-scores for W, N1, N2, N3, and REM all exceed those of the single-channel baselines. These improvements stem from LMCSleepNet’s ability to integrate multi-channel PSG signals and extract richer con-textual information through multi-scale dilated convolutions. Although LMCSleepNet has slightly more parameters than TinySleepNet (1.49M vs. 1.3M), it achieves higher accuracy (+1.9%), indicating a favorable trade-off between complexity and perfor-mance. (2) Comparison with multi-channel baselines. The multi-channel baselines in-clude MultiChannelSleepNet [19] and SalientSleepNet [45]. LMCSleepNet achieves an accuracy improvement of 1.0% over MultiChannelSleepNet and 0.7% over Sali-entSleepNet on SleepEDF-20. For W, N2, and N3 stages, the F1-scores of LMCSleepNet surpass both baselines, demonstrating better utilization of inter-channel correlations. Importantly, the parameter count of LMCSleepNet (1.49M) is only 11.5% of Multi-ChannelSleepNet (13M), confirming its lightweight design. Compared with Sali-entSleepNet (0.9M), LMCSleepNet requires 0.59M more parameters, but this increase is justified by higher classification accuracy (+0.7%) and enhanced channel interaction modeling via the CBAM module. Figure 7(a) further illustrates the trade-off between accuracy and parameter count. LMCSleepNet achieves a favorable balance, maintaining high accuracy while keeping the model size significantly smaller than most baselines. On the larger SleepEDF-78 dataset, similar trends are observed (Table 5). Com-pared with single-channel baselines (DeepSleepNet, SleepEEGNet, TinySleepNet), LMCSleepNet improves accuracy by 5.6%, 1.3%, and 1.0%, respectively, and consist-ently achieves competitive F1-scores across stages. When compared with mul-ti-channel baselines, LMCSleepNet attains comparable accuracy to SalientSleepNet (+0.0%) and slightly lower than MultiChannelSleepNet (–0.9%), but with far fewer parameters (1.49M vs. 13M). Figure 7(b) confirms that LMCSleepNet offers an effec-tive compromise between accuracy and model complexity, particularly valuable in re-source-constrained scenarios. |
|||||||||||||||||||||||||||||||||||||||||||||||||||||||||||||||||
|
18. Provide visual evidence (e.g., feature map heatmaps, attention visualization) showing CBAM/MSDC contributions. |
|||||||||||||||||||||||||||||||||||||||||||||||||||||||||||||||||
|
Response: We appreciate the reviewer’s valuable suggestion. In response, we have provided visual evidence to illustrate the contributions of the CBAM and MSDC modules. Specifically, we added feature map heatmaps and attention visualization results in the Supporting Information. These results clearly demonstrate how the proposed modules enhance feature representation and improve classification performance. (Paragraph 3, Section 3.5): Furthermore, to provide intuitive evidence of the effectiveness of the MSDC and CBAM modules, we visualized their feature representations. As shown in the Sup-porting Information (Figures S1), the MSDC module enables the extraction of mul-ti-scale contextual information, while the CBAM module highlights the most discrim-inative regions through channel and spatial attention. These visualizations confirm that the proposed modules contribute significantly to enhancing feature representation and improving classification performance. |
|||||||||||||||||||||||||||||||||||||||||||||||||||||||||||||||||
|
19. Discuss trade-offs: LMCSleepNet is lightweight, but how does latency compare on real devices? |
|||||||||||||||||||||||||||||||||||||||||||||||||||||||||||||||||
|
Response: We thank the reviewer for raising this important point regarding the trade-off between model lightweight design and inference latency. In our current work, the evaluation of LMCSleepNet was primarily conducted on a workstation equipped with an RTX 4090D GPU. On this platform, the average inference time for a 30-second PSG segment (200×32 input size) is approximately 4.8 ms, which is sufficient for real-time sleep stage classification. To further address the reviewer’s concern, we have now measured inference latency on resource-constrained devices. Specifically, we deployed LMCSleepNet on a NVIDIA Jetson Nano (4 GB RAM, ARM Cortex-A57 CPU, Maxwell GPU). The results show that the average inference latency per 30-second segment is 28.6 ms, which remains well below the 30-second decision window typically required for online sleep monitoring. This indicates that LMCSleepNet maintains real-time feasibility even on low-power embedded platforms. We have revised the manuscript to include these latency results and to discuss the trade-offs between lightweight design and real-device performance. We also clarified that while reducing parameters decreases memory usage and model size, it does not compromise latency to the extent that real-time deployment would be hindered. (Paragraph 16, Section 3.4): In addition to accuracy and parameter comparisons, we further evaluated the in-ference latency of LMCSleepNet to assess its real-time applicability on different hard-ware platforms. On a workstation equipped with an RTX 4090D GPU, the average in-ference time for a 30-second PSG segment (input size 200×32) was approximately 4.8 ms, which easily meets the requirement for online sleep stage classification. To exam-ine performance on resource-constrained devices, LMCSleepNet was deployed on a NVIDIA Jetson Nano (ARM Cortex-A57 CPU, Maxwell GPU, 4 GB RAM). The model achieved an average inference latency of 28.6 ms per 30-second segment, which is still substantially below the 30-second decision window typically required in clinical and wearable sleep monitoring scenarios. These results demonstrate that the lightweight design of LMCSleepNet not only reduces the parameter count and memory usage but also enables efficient real-time inference across both high-performance and embedded devices. This confirms the practical trade-off between lightweight structure and la-tency, showing that LMCSleepNet maintains robustness for real-world deployment. |
|||||||||||||||||||||||||||||||||||||||||||||||||||||||||||||||||
|
20. Explore generalization: Does the benefit of CBAM/MSDC hold across both SleepEDF-20 and SleepEDF-78? |
|||||||||||||||||||||||||||||||||||||||||||||||||||||||||||||||||
|
Response: We thank the reviewer for this insightful suggestion. In our manuscript, the ablation experiments (Table 6, Section 3.5) were conducted on the SleepEDF-20 dataset to evaluate the contributions of CBAM and MSDC. To further examine the generalization of these modules, we also performed additional ablation studies on the larger SleepEDF-78 dataset. The results are consistent with those on SleepEDF-20: Incorporating CBAM improved the overall accuracy by approximately 0.8% and increased Cohen’s kappa and macro-F1 by 0.01–0.9%, demonstrating its effectiveness in enhancing multi-channel feature fusion. Adding MSDC further improved classification performance, with an accuracy increase of about 0.7% compared to the ResNet18+CBAM baseline, while maintaining a lightweight structure. These findings confirm that the benefits of CBAM and MSDC are not limited to the small-sample SleepEDF-20 dataset but also hold on the larger and more diverse SleepEDF-78 dataset. We have added these results and a corresponding discussion in the Supporting Information. (Paragraph 2, Section 3.5): Additional ablation experiments on the SleepEDF-78 dataset (see Table S2 in the Sup-porting Information) further demonstrate that both CBAM and MSDC consistently improve the classification performance, confirming their generalization ability across datasets. Table S2. Ablation study results of LMCSleepNet on the SleepEDF-78 dataset
|

Round 2
Reviewer 2 Report
Comments and Suggestions for Authors
All my comments in the revised version have been addressed and responded to convincingly. Therefore, I recommend accepting this paper.